# GROUNDING PLANNABLE LIFTED ACTION MODELS FOR VISUALLY GROUNDED LOGICAL PREDICATES

## ABSTRACT

We propose FOSAE++, an unsupervised end-to-end neural system that generates a compact discrete state transition model (dynamics / action model) from raw visual observations. Our representation can be exported to Planning Domain Description Language (PDDL), allowing symbolic state-of-the-art classical planners to perform high-level task planning on raw observations. FOSAE++ expresses states and actions in First-Order Logic (FOL), a superset of so-called object-centric representation. It is the first unsupervised neural system that fully supports FOL in PDDL action modeling, while existing systems are limited to continuous, propositional, or property-based representations, and/or require manually labeled input.

## 1 INTRODUCTION

Learning a high-level symbolic transition model of an environment from raw input (e.g., images) is a major challenge in the integration of connectionism and symbolism. Doing so without manually defined symbols is particularly difficult as it requires solving both the Symbol Grounding (Harnad, 1990; Taddeo & Floridi, 2005; Steels, 2008) and the Action Model Learning/Acquisition problem.

Recently, seminal work by Asai & Fukunaga (2018, Latplan) that learns discrete planning models from images has opened the door to applying symbolic Classical Planning systems to a wide variety of raw, noisy data. Latplan uses discrete variational autoencoders to generate propositional latent states and its dynamics (action model) directly from images. Unlike existing work, which requires several machine learning pipelines (SVM/decision trees) and labeled inputs (e.g., a sequence of high-level options) (Konidaris et al., 2014), Latplan is an end-to-end unsupervised neural network that requires no manually labeled inputs. Numerous extensions and enhancements have been proposed: Causal InfoGAN (Kurutach et al., 2018) instead uses GAN framework to obtain propositional representations. Latplan's representation was shown to be compatible with symbolic Goal Recognition (Amado et al., 2018). First-Order State AutoEncoder (Asai, 2019, FOSAE) extends Latplan to generate predicate symbols. Cube-Space AutoEncoder (Asai & Muise, 2020, CSAE) regularized the latent space to a particular form which directly exports to a learned propositional PDDL model (Fikes et al., 1972). Discrete Sequential Application of Words (DSAW) learns a plannable propositional word embedding from a natural language corpus (Asai & Tang, 2020).

In this paper, we obtain a *lifted action model* expressed in First-Order Logic (FOL), which is a superset of *object-centric* (property-based) representation that Machine Learning community recently began to pay attention to[1], but has long been the central focus of the broader AI community. In propositional action models, the environment representation is a fixed-sized binary array and does not transfer to a different or a dynamically changing environment with a varying number of objects. In contrast, lifted FOL representations are generalized over objects and environments, as we demonstrate in Blocksworld with different number of blocks, or Sokoban with different map sizes. We propose *Lifted First-Order Space AutoEncoder* (FOSAE++) neuro-symbolic architecture, which learns a lifted PDDL action model by integrating and extending the FOSAE, the CSAE and the Neural Logic Machine (Dong et al., 2019, NLM) architectures.

The overall task of our system is illustrated in Fig. 1. The system takes a *transition dataset* containing a set of pairs of raw observations which are single time step away. Each observation consists of multiple visual segmentations of the objects. It learns a lifted action model of the environment

---

[1]e.g., ICML workshop on Object-Oriented Learning `https://oolworkshop.github.io/`

by generating the symbols and emits a PDDL (Haslum et al., 2019) encoding for state-of-the-art planning systems.

**Contribution** Table 1 contains a taxonomy of existing model acquisition systems in chronological order. FOSAE++ is the first system that satisfies all features readily available in symbolic action model acquisition systems, while not relying on human-derived symbols. FOSAE++ generates unnamed symbols by itself — Effectively addressing the long-standing Knowledge Acquisition bottleneck (Cullen & Bryman, 1988) and the Symbol Grounding problem, showing a future direction for high-level symbolic autonomy.

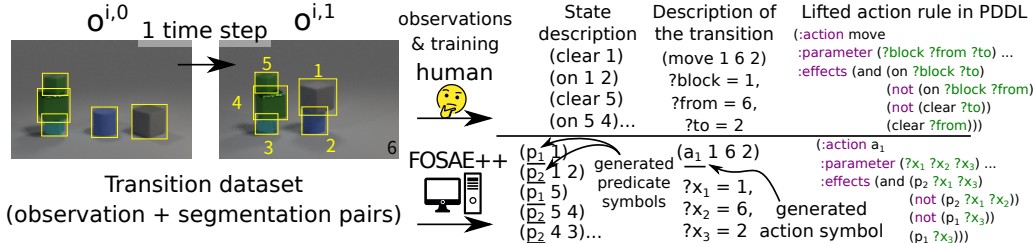

Figure 1: FOSAE++'s learning task compared to the action modeling by humans. All symbols are anonymous (GENSYM'd) due to the unsupervised nature. The produced PDDL files are then used to reason in the environment using classical planners.

| | Symbol Generation | | | Compatibility | | | |
| | Propo-sitional | Multi-arity predicates | Action | End-to-End NN | Search algorithms | Propositional PDDL | Lifted PDDL |
|---|---|---|---|---|---|---|---|
| Symbolic approaches[1] | manual | manual | manual | no | **yes** | **yes** | **yes** |
| (Konidaris et al., 2014) | **yes** | unused | manual[2] | no | **yes** | **yes** | no |
| Latplan (Asai & Fukunaga, 2018) | **yes** | unused | **yes** | **yes** | **yes** | no | no |
| CausalInfoGAN (Kurutach et al., 2018) | **yes** | unused | unused | **yes** | incomplete[2] | no | no |
| FOSAE (Asai, 2019) | **yes** | **yes** | manual | **yes** | **yes** | no | no |
| (James et al., 2020b) | **yes** | incomplete[2] | manual[2] | no | **yes** | **yes** | incomplete[2] |
| Cube-Space AE (Asai & Muise, 2020) | **yes** | unused | **yes** | **yes** | **yes** | **yes** | no |
| **FOSAE++** | **yes** | **yes** | **yes** | **yes** | **yes** | **yes** | **yes** |

Table 1: Taxonomy of action model acquisition systems in chronological order.

## 2 PRELIMINARIES AND BACKGROUND

We denote a multi-dimensional array (tensor) in bold and its elements with a subscript (e.g., $x \in \mathbb{R}^{N \times M}$, $x_2 \in \mathbb{R}^M$), an integer range $n \leq i \leq m$ by $n..m$, a concatenation of tensors $a$ and $b$ in the last axis by $a; b$, and the $i$-th data point of a dataset by a superscript $^i$ which we may omit for clarity. **We use the same symbol for a set and its size (e.g., $S$, and not $|S|$) to avoid the clutter.** Finally, $\mathbb{B} = [0, 1] \subset \mathbb{R}$. We assume background knowledge of discrete VAEs with continuous relaxations (included in the appendix Sec. A.1), such as Gumbel-Softmax (GS) and Binary-Concrete (BC) (Jang et al., 2017; Maddison et al., 2017). Their activations are denoted as GS and BC, respectively.

### 2.1 LIFTED STRIPS/PDDL PLANNING

Planning Domain Description Language (PDDL) is a modeling language for a Lifted STRIPS planning formalism (Fikes et al., 1972) and its extensions (Haslum et al., 2019). Let $\mathcal{F}(T)$ be

---

[1](Yang et al., 2007; Cresswell et al., 2013; Aineto et al., 2018; Zhuo et al., 2019; Cresswell & Gregory, 2011; Mourão et al., 2012; Zhuo & Kambhampati, 2013)

[2]Konidaris et al. (2014) requires sequences of high-level options to learn from, such as [move, move, interact, · · · ] in Playroom domain. Causal InfoGAN cannot deterministically enumerate all successors (a requirement for search completeness) due to the lack of action symbols and should sample the successors. James et al. (2020b)'s PDDL output is limited to unary predicates / properties of objects, thus cannot model the interactions between objects. Also, it requires sequences of high-level options such as [WalkToItem, AttachBlock, WalkToItem, · · · ] in the Minecraft domain.

a formula consisting of logical operations $\{\wedge, \neg\}$ and a set of terms $T$. For example, when $T = \{have(I, food), full(I)\}$, then $have(I, food) \wedge \neg full(I) \in \mathcal{F}(T)$. We denote a lifted STRIPS planning problem as a 5-tuple $\langle O, P, A, I, G \rangle$. $O$ is a set of objects ($\ni food$), $P$ is a set of predicates ($\ni full(x)$), and $A$ is a set of lifted actions ($\ni eat$). Each predicate $p \in P$ has an arity $\#p \geq 0$. Predicates are instantiated/*grounded* into propositions $P(O) = \bigcup_{p \in P} \left( \{p\} \times O \times .^{\#p}. \times O \right)$, such as *have(I, food)*. A state $s \subseteq P(O)$ represents truth assignments to the propositions, e.g., $s = \{have(I, food)\}$ represents *have(I, food)* $= \top$. We can also represent it as a bitvector of size $\sum_p O^{\#p}$.

Each lifted action $a(X) \in A$ has an arity $\#a$ and parameters $X = (x_1, \cdots, x_{\#a})$, such as *eat($x_1, x_2$)*. Lifted actions are instantiated into *ground actions* $A(O) = \bigcup_{a \in A} \left( \{a\} \times O \times .^{\#a}. \times O \right)$, such as *eat(I, food)*. $a(X)$ is a 3-tuple $\langle \text{PRE}(a), \text{ADD}(a), \text{DEL}(a) \rangle$, where $\text{PRE}(a), \text{ADD}(a), \text{DEL}(a) \in \mathcal{F}(P(X))$ are preconditions, add-effects, and delete-effects: e.g., *eat($x_1, x_2$)* $= \langle \{have(x_1, x_2)\}, \{full(x_1)\}, \{have(x_1, x_2)\} \rangle$. The semantics of these three elements are as follows: A ground action $a^\dagger \in A(O)$ is *applicable* when a state $s$ *satisfies* $\text{PRE}(a^\dagger)$, i.e., $\text{PRE}(a^\dagger) \subseteq s$, and applying an action $a^\dagger$ to $s$ yields a new successor state $a^\dagger(s) = (s \setminus \text{DEL}(a^\dagger)) \cup \text{ADD}(a^\dagger)$, e.g., *eat(I, food)* $=$ "*I can eat a food when I have one, and if I eat one I am full but the food is gone.*" Finally, $I, G \subseteq P(O)$ are the initial state and a goal condition, respectively. The task of classical planning is to find a *plan* $(a_1^\dagger, \cdots, a_n^\dagger)$ which satisfies $a_n^\dagger \circ \cdots \circ a_1^\dagger(I) \subseteq G$.

## 2.2 NEURAL PROPOSITIONAL/ACTION SYMBOL GENERATION WITH LATPLAN

Latplan is a framework for *domain-independent image-based classical planning* (Asai & Fukunaga, 2018). It learns a propositional state representation and transition rules entirely from image-based observations of the environment with discrete VAEs and solves the problem using a classical planner. Latplan is trained on a *transition input* Tr: a set of pairs of raw data randomly sampled from the environment. The $i$-th transition $(o^{i,0}, o^{i,1}) \in$ Tr is a pair of observations made before and after an unknown high-level action is performed. Once trained, Latplan can process a *planning input* $(o^I, o^G)$, a pair of raw images corresponding to an initial and goal state of the environment. The output of Latplan is a data sequence representing the plan execution $(o^I, \ldots o^G)$ that reaches $o^G$ from $o^I$. While the original paper used an image-based implementation, conceptually any form of temporal data is viable for this methodology, e.g., an NLP corpus (Asai & Tang, 2020).

The latest Latplan Asai & Muise (2020) has a training phase and a planning phase. In the training phase, it trains an end-to-end neural network called *Cube-Space AutoEncoder* (CSAE) on Tr (Fig. 2, top left). CSAE is a variational autoencoder modeled by binary and categorical random variables each representing the propositional states and the actions in the classical planning. The dynamics modeled by these actions directly compiles into a PDDL model. The network combines Binary-Concrete VAE to produce binary state representation, and a Gumbel-Softmax VAE to produce a

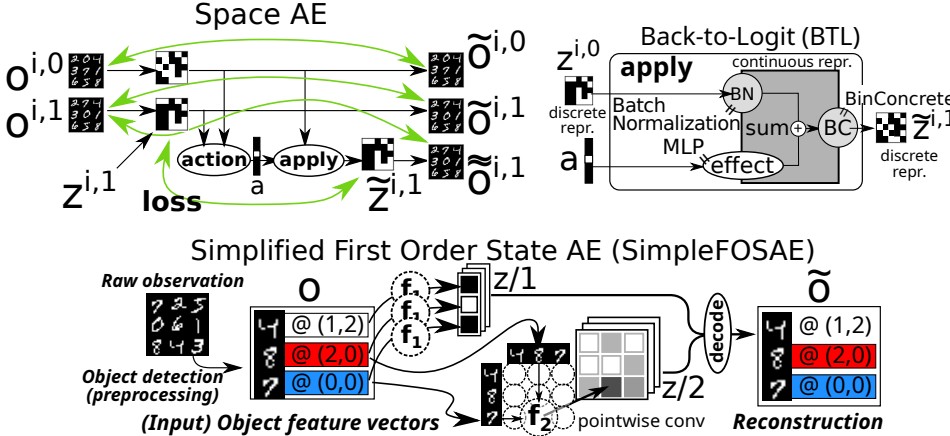

Figure 2: Cube-Space AutoEncoder, Back-To-Logit, and First-Order State AutoEncoder.

categorical bottleneck layer which assigns a categorical label to each input. Let $\boldsymbol{o}^0$ and $\boldsymbol{o}^1$ be a pair of observed states in a transition, $\boldsymbol{z}^0$ and $\boldsymbol{z}^1$ be the corresponding binary latent states, and $\boldsymbol{a}$ be the one-hot vector that represents a discrete action label assigned to the transition. CSAE is a variational autoencoder network that can be formalized as follows:

$$\text{(encoder)} \qquad \boldsymbol{z}^0, \boldsymbol{z}^1 = \text{ENCODE}(\boldsymbol{o}^0), \text{ENCODE}(\boldsymbol{o}^1) \in \mathbb{B}^F$$

$$\text{(action assignment/clustering)} \qquad \boldsymbol{a} = \text{ACTION}(\boldsymbol{z}^0, \boldsymbol{z}^1) \in \mathbb{B}^A$$

$$\text{(learning the dynamics)} \qquad \tilde{\boldsymbol{z}}^1 = \text{APPLY}(\boldsymbol{z}^0, \boldsymbol{a}) \in \mathbb{B}^F$$

$$\text{(decoder)} \qquad \tilde{\boldsymbol{o}}^0, \tilde{\boldsymbol{o}}^1, \tilde{\tilde{\boldsymbol{o}}}^1 = \text{DECODE}(\boldsymbol{z}^0), \text{DECODE}(\boldsymbol{z}^1), \text{DECODE}(\tilde{\boldsymbol{z}}^1)$$

where ENCODE, DECODE, ACTION, APPLY are arbitrary multilayer perceptrons. The outputs of ENCODE and APPLY are activated by Binary Concerete, and the output of ACTION is activated by Gumbel-Softmax of $A$ categories (hyperparameter). Assuming a certain set of prior distributions, a lower bound (ELBO) of the log likelihood of observing a pair of states $(\boldsymbol{o}^0, \boldsymbol{o}^1)$ can be as derived follows (Appendix Sec. A.3.3):

$$\log p(\boldsymbol{o}^0, \boldsymbol{o}^1) \geq -D_{\text{KL}}(q(\boldsymbol{a}|\boldsymbol{o}^1, \boldsymbol{z}^0, \boldsymbol{z}^1)||p(\boldsymbol{a}|\boldsymbol{z}^0, \boldsymbol{z}^1)) - D_{\text{KL}}(q(\tilde{\boldsymbol{z}}^1|\boldsymbol{o}^1, \boldsymbol{a}, \boldsymbol{z}^0, \boldsymbol{z}^1)||q(\boldsymbol{z}^1|\boldsymbol{o}^1))$$

$$-D_{\text{KL}}(q(\boldsymbol{z}^0|\boldsymbol{o}^0)||p(\boldsymbol{z}^0)) - D_{\text{KL}}(q(\boldsymbol{z}^1|\boldsymbol{o}^1)||p(\boldsymbol{z}^1)) + \log p(\boldsymbol{o}^0|\boldsymbol{z}^0) + \log p(\boldsymbol{o}^1|\tilde{\boldsymbol{z}}^1, \boldsymbol{a}, \boldsymbol{z}^0, \boldsymbol{z}^1)$$

After the training, it generates a propositional classical planning problem $(\boldsymbol{z}^I, \boldsymbol{z}^G)$ from a planning input $(\boldsymbol{o}^I, \boldsymbol{o}^G)$ and export it into PDDL files together with the action model obtained in (2), which are then solved by Fast Downward (Helmert, 2006), an optimized C++-based solver independent from the neural network. Finally, it obtains a step-by-step, human-comprehensible visualization of the plan execution by decoding the intermediate states of the plan into images.

The $A$-dimensional one-hot categorical variable $\boldsymbol{a}^i$ in the network performs a clustering on the state transitions with the maximum number of clusters $A$ specified as a hyperparameter. The cluster ID is used as its action symbol. The clustering is performed by its encoder, $\text{ACTION}(\boldsymbol{z}^{i,0}, \boldsymbol{z}^{i,1}) = \boldsymbol{a}^i$, which takes a propositional state pair $(\boldsymbol{z}^{i,0}, \boldsymbol{z}^{i,1})$ and returns a one-hot vector $\boldsymbol{a}^i$ of $A$ categories using Gumbel-Softmax. AAE's decoder APPLY takes the current state $\boldsymbol{z}^{i,0}$ and the action $\boldsymbol{a}^i$ and reconstructs a successor state $\tilde{\boldsymbol{z}}^{i,1} \approx \boldsymbol{z}^{i,1}$, acting as a progression function $\text{APPLY}(\boldsymbol{a}^i, \boldsymbol{z}^{i,0}) = \tilde{\boldsymbol{z}}^{i,1}$. APPLY is typically just called a "model" in model-based RL literature.

While APPLY can be any network from the training standpoint, such a neural black-box function does not directly translates to a STRIPS action model, preventing efficient search with state-of-the-art classical planner. Cube-Space AE (Asai & Muise, 2020) addresses this issue by Back-To-Logit technique (BTL Fig. 2, bottom-left) which modifies APPLY. Latent state transitions learned by BTL guarantees that the actions and the transitions satisfy the STRIPS state transition rule $s' = (s \setminus \text{DEL}(a)) \cup \text{ADD}(a)$, thus enabling a direct translation from neural network weights to PDDL modeling language. Details of the network, the translation method and the proof can be found in the appendix Sec. A.3.

## 2.3 PREDICATE SYMBOL GENERATION WITH SIMPLE FIRST-ORDER STATE AUTOENCODER

First-Order State AutoEncoder (Asai, 2019, FOSAE, Fig. 2, bottom) is an autoencoder that takes a set of *object feature vectors*, identifies its latent predicate representation, then outputs a reconstruction of the input. Unlike prior work on relational modeling (Santoro et al., 2017; Zambaldi et al., 2019; Battaglia et al., 2018), this system obtains discrete logical predicates compatible with symbolic systems. The input is similar to the setting of Ugur & Piater (2015) and James et al. (2020b): Each feature vector can represent each object in the environment in an arbitrary complex manner. In this paper, we use segmented pixels combined with bounding box information ($x, y$ and width, height). Let $\boldsymbol{o}_n \in \mathbb{R}^F$ be an $F$-dimensional feature vector representing each object and $\boldsymbol{o} = (\boldsymbol{o}_1, \ldots \boldsymbol{o}_O) \in \mathbb{R}^{O \times F}$ be the input matrix representing the set of $O$ objects.

FOSAE generates a *multi-arity latent representation*. Assume we represent the input with a combination of predicates of different arities. We denote a set of predicates of arity $n$ as $P/n$ (Prolog notation) and its propositions as $P/n(O)$. We denote the binary tensor representation of $P/n(O)$ as $\boldsymbol{z}/n \in \mathbb{B}^{O \times \overset{n}{\ldots} \times O \times P/n}$, and the latent space is a tuple $\boldsymbol{z} = (\boldsymbol{z}/1, \ldots, \boldsymbol{z}/N)$, where $N$ is the largest arity. The total size of its latent space is $\sum_{n=1}^{n=N} O^n P/n$. To convert the input $\boldsymbol{o}$ into unary

predicates $P/1$, we apply a 1D pointwise convolutional filter $f_1$ of $P/1$ output features over the objects and activate it by BinaryConcrete for discretization, i.e., $z/1_i = \text{BC}(f_1(o_i)) \in \mathbb{B}^{P/1}$. Similarly, for binary relationships $P/2$, we apply a filter $f_2$ over concatenated pairs of objects, i.e., $z/2_{ij} = \text{BC}(f_2(o_i; o_j)) \in \mathbb{B}^{P/2}$. We can extend this framework to an arbitrary arity $n$. The multi-arity latent tensors $(z/1, \ldots, z/N)$ can be flattened, concatenated, then decoded to the reconstruction $\tilde{o} \in \mathbb{R}^{O \times F}$ by an arbitrary decoder network (MLP in the original paper).

There is a potential exponential blowup due to $O^N$ parameter combinations. This problem is addressed by *attentions* in the original work (Asai, 2019), but we omit this feature in this paper for simplicity and due to relatively small $N (< 4)$. We call our variant as SimpleFOSAE.

## 2.4 NEURAL LOGIC MACHINE (NLM)

The NLM (Dong et al., 2019) is a neural Inductive Logic Programming (ILP) (Muggleton, 1991) system whose inputs and outputs are hand-crafted binary tensors representing propositional groundings of FOL statements. These binary tensors are in the multi-arity representation $(z/1, \ldots, z/N)$ identical to the latent space of SimpleFOSAE. NLM provides a key feature that is necessary for learning a lifted First-Order Logic action model: Invariance to the number and the order of objects in the multi-arity representation. The latter implies permutation equivariance, i.e., when the first $n$ axes of each input $z/n \in \mathbb{B}^{O \times \overset{n}{\cdots} \times O \times P/n}$ are reordered by a permutation $\pi$, the output axes are also reordered by $\pi$. A more detailed explanation of NLM can be found in the appendix Sec. A.2.

An NLM contains $N$ COMPOSE operations which are applied to the corresponding elements of $(z/1, \cdots, z/N)$. Each COMPOSE is denoted as $\text{COMPOSE}_{Q,\sigma}(z, n) \in \mathbb{B}^{O \cdots O \times Q}$, where $Q$ is a constant that specifies the number of output features, and $\sigma$ is a nonlinearlity. An NLM output is denoted as $\text{NLM}_{Q,\sigma}(z) = (\text{COMPOSE}_{Q,\sigma}(z, 1), \cdots, \text{COMPOSE}_{Q,\sigma}(z, N))$.

## 3 LIFTING THE ACTION MODEL

In order to obtain a lifted action model in an unsupervised setting, there are three requirements: (1) white-box action model which is trivially convertible to STRIPS formalism, (2) invariance to the number/order of objects, (3) unsupervised generation of multi-arity predicate symbols. To our knowledge, no existing systems fulfill all requirements: (Simple)FOSAE lacks 1 and 2, CSAE lacks 2 and 3, and NLM lacks 1 and 3 (designed for hand-crafted symbolic data).

$$(\text{FOSAE encoder}) \quad z^{i,0}, z^{i,1} = \text{ENCODE}(o^{i,0}), \text{ENCODE}(o^{i,1})$$

$$\textbf{(action parameters selector in NLM)} \quad x^i = \text{PARAMS}(z^{i,0}, z^{i,1})$$

$$\textbf{(extract the parameter-bound subspace)} \quad z_{\dagger}^{i,0}, z_{\dagger}^{i,1} = \text{BIND}(z^{i,0}, x^i), \text{BIND}(z^{i,1}, x^i)$$

$$(\text{unsupervised action assignment}) \quad a^i = \text{ACTION}(z_{\dagger}^{i,0}, z_{\dagger}^{i,1})$$

$$(\text{learning the bounded dynamics}) \quad \tilde{z}_{\dagger}^{i,1} = \text{APPLY}(z_{\dagger}^{i,0}, a^i)$$

$$\textbf{(reflection to the global dynamics)} \quad \tilde{z}^{i,1} = z^{i,0} - \text{UNBIND}(z_{\dagger}^{i,0}, x^i) + \text{UNBIND}(\tilde{z}_{\dagger}^{i,1}, x^i)$$

$$(\text{decoder in NLM}) \quad \tilde{o}^{i,0}, \tilde{o}^{i,1}, \tilde{\tilde{o}}^{i,1} = \text{DECODE}(z^{i,0}), \text{DECODE}(z^{i,1}), \text{DECODE}(\tilde{z}^{i,1})$$

$$\text{Loss: } \ell(o^{i,0}, \tilde{o}^{i,0}) + \ell(o^{i,1}, \tilde{o}^{i,1}) \quad + \ell(o^{i,1}, \tilde{\tilde{o}}^{i,1}) + \ell(z^{i,1}, \tilde{z}^{i,1}) + \ell(z_{\dagger}^{i,1}, \tilde{z}_{\dagger}^{i,1}) + \text{regularization.}$$

We now propose FOSAE++ whose overview is shown above, which addresses all issues at once. Its overall design follows the CSAE, which ENCODE s the input, identifies the action $a^i$, APPLY-es the action, then DECODE s the results. It uses FOSAE's encoder to generate multi-arity latent representation, which is then consumed by NLMs used as a basic building block of other components. *This architecture's key element is a new component* PARAMS *and a unique pair of operations called* BIND *and* UNBIND, *which intuitively reflects the structure of lifted action models.* Suppose we model a lifted action (move ?block ?from ?to) in Blocksworld (Fig. 1), with effects such as (on ?block ?to). Since a lifted model always refers to the objects through its parameters such as ?to, it cannot affect

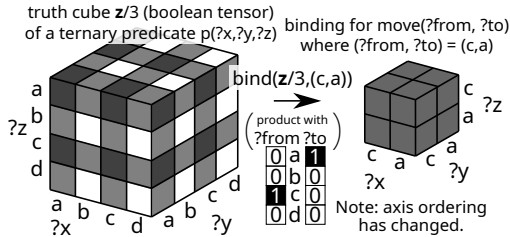

Figure 3: move(c,a) should not affect p's groundings with b and d, therefore BIND limits the grounding to a and c. move(c,a) and move(a,c) can then apply the same effect to each subspace due to the axis reordering. (Note: move(?block, ?from,?to) is simplified to move(?from,?to) for illustration.)

the objects that does not appear in the parameter list, e.g., action (move a b c) cannot affect (on d c) because $d \notin \{a, b, c\}$. *We represent this restriction as differentiable matrix operations.*

To implement this idea, we first added a new attention network $\boldsymbol{x} = \text{PARAMS}(\boldsymbol{z}^{i,0}; \boldsymbol{z}^{i,1}) = (\boldsymbol{x}_1, \ldots, \boldsymbol{x}_{\#a})$ which attends to the objects, essentially learning the parameters of the action. We assume all actions have the same number of parameters, and thus $\#a$ is a hyperparameter. Each parameter $\boldsymbol{x}_i$ is a one-hot hard attention vector in $\mathbb{B}^O$ ($\sum_{j=1}^{j=O} \boldsymbol{x}_{ij} = 1$) activated by Gumbel-Softmax. This can later extract a specific object from the object feature matrix $\boldsymbol{o}$ by an inner product $\boldsymbol{x}_i \cdot \boldsymbol{o}$ ($\mathbb{B}^O \mathbb{R}^{O \times F} \to \mathbb{R}^F$). PARAMS has several NLM layers, ending with $\text{NLM}_{\#a, \text{IDENTITY}}$. We then extract its unary part of the results ($\to \mathbb{R}^{O \times \#a}$), transpose it ($\to \mathbb{R}^{\#a \times O}$), then apply $\#a$-way Gumbel-Softmax of $O$ categories ($\to \mathbb{B}^{\#a \times O}$). As a result, the output attends to $\#a$ objects in total.

We utilize the attentions in unique operations named BIND and UNBIND. Recall that the predicates in the effects can refer only to the specified action parameters. Therefore, we *limit the dynamics to the objects attended by $\boldsymbol{x}$* (Fig. 3). We iteratively extracts the sub-axes of $\boldsymbol{z}/n$ attended by $\boldsymbol{x}$ using matrix operations (with an abuse of notation) $\boldsymbol{z}_\dagger/n = (\boldsymbol{x})^n \boldsymbol{z}/n$ (($\mathbb{B}^{\#a \times O})^n \mathbb{B}^{O.\overset{n}{\ldots}.O \times P/n} \to \mathbb{B}^{\#a.\overset{n}{\ldots}.\#a \times P/n}$). We call it $\text{BIND}(\boldsymbol{z}/n, \boldsymbol{x})$, as it *binds* the parameters $X$ to the values in objects $O$, similar to the function call in a typical programming language. It is also similar to applying numpy.take in all dimensions. Binding $\boldsymbol{z}/n$ for all arities results in $\text{BIND}(\boldsymbol{z}, \boldsymbol{x}) = (\text{BIND}(\boldsymbol{z}/1, \boldsymbol{x}), \cdots \text{BIND}(\boldsymbol{z}/N, \boldsymbol{x}))$. We also define $\text{UNBIND}(\boldsymbol{z}_\dagger/n, \boldsymbol{x}) = (\boldsymbol{x}^\top)^n \boldsymbol{z}_\dagger/n \in \mathbb{B}^{O.\overset{n}{\ldots}.O \times P/n}$ which restores the original shape, but fills the cells with zeros for the objects not attended by $\boldsymbol{x}$.

**Example 1.** *For a moment let's ignore that we use a boolean tensor and let $\boldsymbol{z}/2 = [[1, 2, 3], [4, 5, 6], [7, 8, 9]]$ with $O = \{1, 2, 3\}$ which represents a two-arg function $f(2, 1) = \boldsymbol{z}/2_{2,1} = 4$ etc. When we bind $\boldsymbol{z}/2$ to $(2, 1)$ by $\boldsymbol{x} = [[.01, .98, .01], [.98, .01, .01]]$, we obtain $\boldsymbol{z}_\dagger/2 = \boldsymbol{x}(\boldsymbol{z}/2)\boldsymbol{x}^\top \approx [[5., 4.], [2.1, 1.1]] \approx [[5, 4], [2, 1]]$, approximating the subspace extraction. To unbind it, $\boldsymbol{x}^\top(\boldsymbol{z}_\dagger/2)\boldsymbol{x} \approx [[1.1, 2.1, .03], [3.9, 4.9, .09], [.05, .07, .01]] \approx [[1, 2, 0], [4, 5, 0], [0, 0, 0]]$.*

After the subspace extraction with BIND, we use APPLY and ACTION to find the dynamics / action model inside the subspace. Notice that ACTION and APPLY no longer directly refers to each object $\boldsymbol{o}_i$ by the index $i$ because BIND already resolves the index reference. They take flattened binary vectors of size $\sum_n (\#a)^n P/n$ as the input / output.

In order to reflect the changes applied to the bounded representation to the whole representation, we use UNBIND: $\tilde{\boldsymbol{z}}^{i,1} = \boldsymbol{z}^{i,0} - \text{UNBIND}(\boldsymbol{z}_\dagger^{i,0}) + \text{UNBIND}(\tilde{\boldsymbol{z}}_\dagger^{i,1})$. UNBIND is unique in that the unattended axes have near-0 values (0 at the limit of Gumbel-Softmax annealing in PARAMS). Therefore, the propositions not bound by $\boldsymbol{x}$ retain their values from $\boldsymbol{z}^{i,0}$.

Finally, we replace SimpleFOSAE's MLP decoder with NLM layers. To match the output shape with the input $\boldsymbol{o} \in \mathbb{R}^{O \times F}$, the final layer has $F$ features and we use only the unary part of the result tuple, $\text{NLM}_{F,\sigma}(\cdot)/1 = \text{COMPOSE}(\cdot, 1, F, \sigma)$. The total loss is the sum of $\ell(\boldsymbol{o}^{i,0}, \tilde{\boldsymbol{o}}^{i,0})$, $\ell(\boldsymbol{o}^{i,1}, \tilde{\boldsymbol{o}}^{i,1})$, $\ell(\boldsymbol{o}^{i,1}, \tilde{\overset{\approx}{\boldsymbol{o}}}^{i,1})$, $\ell(\boldsymbol{z}^{i,1}, \tilde{\boldsymbol{z}}^{i,1})$, and $\ell(\boldsymbol{z}_\dagger^{i,1}, \tilde{\boldsymbol{z}}_\dagger^{i,1})$, plus the variational losses for discrete VAEs. Hyperparameters, choice of loss functions and the tuning process are detailed in the appendix Sec. C.

The number of parameters in FOSAE++ is not affected by the number of objects, since essentially each NLM layer performs a 1D pointwise convolution over tuples of objects.

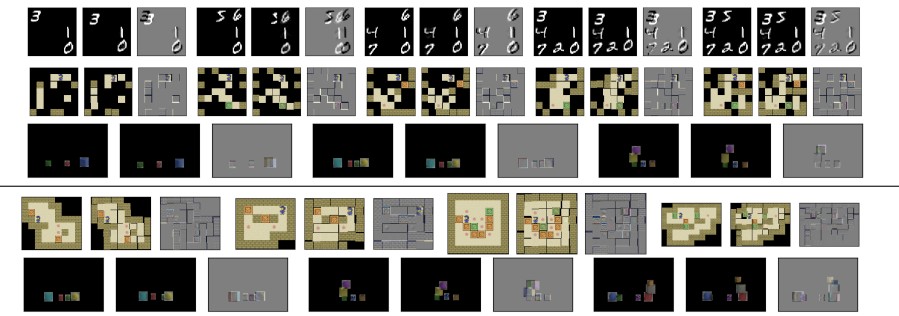

Figure 4: (**top**) State transitions in Blocksworld and 8-puzzle, and the initial/goal state of a Sokoban problem. In Sokoban and 8-puzzle, each tile is represented as an object. (**bottom**) Example rendering of the ground-truth / reconstruction / errors. FOSAE++ reconstructs rectangular images and bounding boxes, then draw them on a black canvas. Due to the object-based input, it does not produce the background. The depicted Sokoban reconstruction consists of 63 tile objects.

Figure 5: (**Synopses:**) The networks are trained on $X$ objects, then tested on $Y$ objects. (**Top: Interpolation**) 8-Puzzle: $X = 9$, $Y \in \{3, 4, 5, 6, 7\}$. Sokoban: $X = 46$, $Y \in \{13, 18, 23, 27, 32\}$. Blocksworld: $X = 6$, $Y \in \{3, 4, 5\}$. (**Bottom: Extrapolation**) Sokoban: $X = 46$ (problem p0), $Y \in \{64, 41, 64, 43\}$ (p01-p04). Blocksworld: $X = 3$, $Y \in \{4, 5, 6\}$.

## 4 EXPERIMENTS

We prepared 3 artificial environments: **Photo-realistic Blocksworld** (Asai, 2018) renders an RGB Blocksworld image with Blender 3D engine and extracts the bounding boxes and the image patches of each object. **MNIST 8-Puzzle** (Asai & Fukunaga, 2018) is a 42x42 pixel, monochrome image-based version of the 8-Puzzle. Tiles contain hand-written digits (0-9) from the MNIST database (LeCun et al., 1998), and valid moves swap the "0" tile with a neighboring tile, i.e., the "0" serves as the "blank" tile in the classic 8-Puzzle. **Sokoban** (Culberson, 1998) is a PSPACE-hard puzzle domain whose visualization was obtained from the PDDLGym library (Silver & Chitnis, 2020). In all datasets, segmentation is already provided by a domain-specific code, but in principle it can be replaced by the output of object-detectors such as YOLO (Redmon & Farhadi, 2018). The detail of the data preparation is available in the appendix Sec. D. Fig. 4 shows some comparisons between the input $o^{i,0}$ and the reconstruction $\tilde{o}^{i,0}$. In Sokoban, FOSAE++ generates up to 63 objects in a single scene.

### 4.1 INTERPOLATION/EXTRAPOLATION TO VARYING, UNSEEN NUMBER OF OBJECTS

The key characteristics of the FOSAE++ is that once trained, its network structurally generalizes to the different number of objects. We demonstrate this performance in *interpolation/extrapolation* tasks, where in the former we randomly drop a certain number of objects from the input, and in the latter, we use an environment with a different distribution or more objects. In both tasks, FOSAE++ shows an excellent reconstruction for a varying, untrained number of objects (Fig. 5).

### 4.2 PLANNING EXPERIMENTS

Finally, we ran Fast Downward Helmert (2006) planning system with LMcut Helmert & Domshlak (2009) heuristics and $A^*$ search configuration on the PDDL domains generated by FOSAE++. For the planning to succeed, FOSAE++ must be accurate not only about the direct reconstruction

$\widetilde{\boldsymbol{o}}^{i,0}, \widetilde{\boldsymbol{o}}^{i,1}$ but also about the dynamics that predicts the successor state $\widetilde{\boldsymbol{z}}^{i,1}$ and its reconstruction $\widetilde{\widetilde{\boldsymbol{o}}}^{i,1}$.

Due to the time constraint, we only performed the experiments for 8-Puzzle as of now. From the fixed goal state (the solved state of the puzzle), we performed a domain-specific Dijkstra search and sampled the initial states optimally $L$-steps away. Each initial/goal states are rendered, cropped into object-based input, then encoded by FOSAE++ to produce the symbolic initial/goal states. The planning results are visualized and manually inspected/validated by us. The symbolic classical planner correctly produced a solution which can be visualized into correct state transitions. See Sec. E.1 for the visualization of all 20 instances.

| 8-Puzzle | | |
|---|---|---|
| $(L = 5)$ | 10/10 |  |
| $(L = 10)$ | 10/10 | |

Table 2: Planning experiments (solved instances / total) and an example visualization obtained from the system ($L = 10$).

## 5   RELATED WORK

James et al. (2020a;b) build on existing work (Konidaris et al., 2014; 2015; 2018) to find an *ego-centric* logical representation that is invariant to the state of the observer in a complex environment, or an *object-centric*, property-based logical representation. Both representations are special cases of First-Order Logic representation where all predicates are unary. These approaches also require a training input that already contains action symbols (e.g., toggle-door). Andersen & Konidaris (2017) obtains an MDP-based PPDDL (Probabilistic PDDL) model using Active Learning. Incorporating Active Learning in FOSAE++ for self data collection is future work.

Huang et al. (2019) reported that direct discretization approach performed worse than an approach that plans in the probabilistic representation obtained by neural networks. However, the question of continuous versus discrete is not conclusive. They used a naive rounding-based discretization which may cause a significant amount of information loss compared to the state-of-the-art discrete variational autoencoders. Furthermore, they rely on mappings from raw inputs to hand-coded symbolic representations that require supervised training.

Object-based input we used can be obtained from state-of-the-art object-recognition methods such as YOLO (Redmon & Farhadi, 2018). More recent systems (Greff et al., 2019; Engelcke et al., 2020) can handle shapes not limited to rectangles, and can be trained unsupervised. Connecting our network with these systems is an exciting avenue of future work.

Eslami et al. (2016) and Xu et al. (2019) proposed an attention-based and a Bayesian approach to the problem of variable-sized object reconstruction. Their work do not address the state dynamics and is orthogonal to our work. The primary difference from our approach is that they studied on sequentially storing and retrieving a variable amount of information into/from a fixed-sized array, while we store them in a latent representation of the corresponding size.

## 6   CONCLUSION

We proposed a first fully neural First-Order Logic Action Model Acquisition system that successfully produces a lifted PDDL planning model from noisy, segmented images alone. The system generates three types of symbols (action, predicate, and propositional) without manual labeling, and provides STRIPS-compatible explanations of actions. The learned results are generalized over the objects, thus can extrapolate to different and dynamically changing environments. We demonstrated that the state-of-the-art classical planner can produce correct solutions (as theoretically guaranteed) only with the interactions with the learned model. Note that the classical planner we used has no learning components. Our results partially support the Physical Symbol Systems Hypothesis by Newell & Simon (1976): In our word, *a formal high-level representation learned from the raw inputs is sufficient for intelligent actions*. Future work includes extensions to the formalisms with higher expressivity, including Probabilistic PDDL (Younes & Littman, 2004), Axioms (Thiébaux

et al., 2005), ADL (Pednault, 1987), numeric planning (Fox & Long, 2006), or Hierarchical Task Networks (Erol et al., 1994; Nau et al., 2003; Nejati et al., 2006).

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

## A   EXTENDED BACKGROUNDS

### A.1   DISCRETE VARIATIONAL AUTOENCODERS

Variational AutoEncoder (VAE) is a framework for reconstructing an observation $x$ from a compact latent representation $z$ that follows a certain prior distribution. Training is performed by maximizing the sum of the reconstruction loss and the KL divergence between the latent random distribution $q(z|x)$ and the target distribution $p(z)$, which gives a lower bound for the likelihood $p(x)$ (Kingma & Welling, 2013). While typically $p(z)$ is a Normal distribution $\mathcal{N}(0,1)$, Gumbel-Softmax (GS) VAE (Jang et al., 2017) and Binary-Concrete (BC) VAE (Maddison et al., 2017) use a discrete, uniform categorical/Bernoulli(0.5) distribution, and further approximate it with a continuous relaxation by introducing a temperature parameter $\tau$ that is annealed down to 0.

We denote corresponding activation function in the latent space as GS($l$) and BC($l$), where $l$ and $l$ each represents a logit vector and scalar. A latent vector $z \in [0,1]^C$ of GS-VAE is computed from a logit vector $l \in \mathbb{R}^C$ by $z = \text{GS}(l) = \text{SOFTMAX}((l + \text{GUMBEL}(0,1))/\tau)$, where $C$ is the number of categories, $\text{GUMBEL}(0,1) = -\log(-\log u)$ and $u \sim \text{UNIFORM}(0,1) \in [0,1]^C$.

A latent scalar $z$ of BC-VAE is computed from a logit scalar $l \in \mathbb{R}$ by $z = \text{BC}(l) = \text{SIGMOID}((l + \text{LOGISTIC}(0,1))/\tau)$, where $\text{LOGISTIC}(0,1) = \log u - \log(1-u)$ and $u \sim \text{UNIFORM}(0,1) \in \mathbb{R}$.

Both functions converge to discrete functions at the limit $\tau \to 0$: $\text{GS}(l) \to \arg\max(l)$ (in one-hot representation) and $\text{BC}(l) \to \text{STEP}(l) = (l < 0)\,?\,0:1$. Typically, GS-VAE and BC-VAE contains multiple latent vectors / latent scalars to model the complex input.

There are more complex variations such as VQVAE van den Oord et al. (2017), DVAE++Vahdat et al. (2018b), DVAE# Vahdat et al. (2018a). They may contribute to the stable performance, but we leave the task of faster / easier training to the future work.

### A.2   NEURAL LOGIC MACHINE

The NLM (Dong et al., 2019) is a neural Inductive Logic Programming (ILP) (Muggleton, 1991) system based on First-Order Logic and the Closed-World Assumption (Reiter, 1981). Given a set of base predicates grounded on a set of objects, NLMs sequentially apply horn rules to draw further conclusions such as a property of or a relationship between objects. For example, in Blocksworld, based on a premise such as (on a b) for blocks a, b, NLMs can infer (not (clear b)). NLM has two unique features: (1) The ability to combine the predicates of different arities, and (2) size invariance & permutation equivaliance on input objects, which is achieved by enumerating the permutations of input arguments.

NLM is designed to work on hand-crafted binary tensors representing propositional groundings of FOL statements. The format of these binary tensors are the multi-arity representation $(z/1, \ldots, z/N)$ identical to the latent space of SimpleFOSAE.

NLM is designed for a subset of First Order Logic where every statement is a horn rule, contains no function terms (such as a function that returns an object), and all rules are applied between neighboring arities. With these assumptions, the statements fall in one of the three types below:

$$
\begin{aligned}
\text{(expand)} \quad & \forall x_{\#\overline{p}}; \overline{p}(X; x_{\#\overline{p}}) \leftarrow p(X), \\
\text{(reduce)} \quad & \underline{p}(X) \leftarrow \exists x_{\#p}; p(X; x_{\#p}), \\
\text{(compose)} \quad & q(X) \leftarrow \mathcal{F}\left(\bigcup_{\pi}(P \cup \underline{P} \cup \overline{P})/_{\#q}(\pi(X))\right).
\end{aligned}
$$

Here, $p, \underline{p}, \overline{p}, q \in P, \underline{P}, \overline{P}, Q$ (respectively) are predicates, $X = (x_1, \ldots)$ is a sequence of parameters, $\mathcal{F}(T)$ is a formula consisting of $\{\wedge, \vee, \neg, \top, \bot\}$ and $T$, $(P \cup \underline{P} \cup \overline{P})/_{\#q}$ is a set of predicates of the same arity as $q$, and $\pi(X)$ is a permutation of $X$, which is used for composing the predicates with the different argument orders.

All three rules can be implemented as tensor operations (Fig. 6). Given a binary tensor $z/n$ of shape $O.^n.O \times P/n$, *expand* copies the $n$-th axis to $n + 1$-th axis resulting in a shape $O_.^{n+1}O \times P/n$, and *reduce* takes the max of $n$-th axis resulting in a shape $O_.^{n-1}O \times P/n$. While the original paper

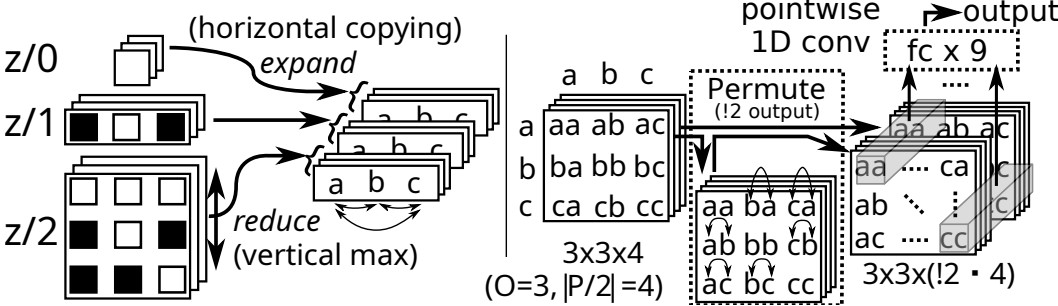

Figure 6: NLM uses EXPAND, REDUCE, PERMUTE tensor operations to combine predicates of different arities while maintaining the object-ordering / size invariance. (For matrices, PERMUTE is equivalent to transposition.)

proposed both $\min$ and $\max$ versions to represent $\forall$ and $\exists$, with enough number of layers only one of them is necessary because $\forall x; p(\cdot, x) = \neg \exists x; \neg p(\cdot, x)$.

Finally the COMPOSE combines the information between the neiboring tensors $z/n, z/_{n-1}, z/_{n+1}$. In order to use the information in the neighboring arities ($P$, $\underline{P}$ and $\overline{P}$), the input concatenates $z/n$ with EXPAND($z/_{n-1}$) and REDUCE($z/_{n+1}$) ($\rightarrow O.^n.O \times C$ where $C = P/n + P/_{n-1} + P/_{n+1}$). Next, a PERMUTE function enumerates and concatenates the results of swapping the first $n$ axes in the tensor ($\rightarrow O.^n.O \times (!n \cdot C)$). It then applies a $n$-D pointwise convolutional filter $f_n$ with $Q$ output features ($\rightarrow O.^n.O \times Q$). In the actual implementation, this $n$-D pointwise filter is merely a 1D convolution performed on a matrix reshaped into $O^n \times (!n \cdot C)$. It is activated by any nonlinearity $\sigma$ to obtain the final result, which we denote as COMPOSE($z, n, Q, \sigma$). Formally, $\forall j \in 1..n, \forall o_j \in 1..O$,

$$\text{COMPOSE}(z, n, Q, \sigma)_{o_1 \cdots o_n} = \sigma(f_n(\Pi_{o_1 \cdots o_n})) \in \mathbb{R}^Q$$

$$\text{where} \quad \Pi = \text{PERMUTE}\Big(\text{EXPAND}(z/_{n-1});\ z/n;\ \text{REDUCE}(z/_{n+1})\Big) \in \mathbb{B}^{O.^n.O \times (!n \cdot C)}$$

An NLM contains $N$ (the maximum arity) COMPOSE operation for the neighboring arities, with appropriately omitting both ends (0 and $N + 1$) from the concatenation. We denote the result as $\text{NLM}_{Q,\sigma}(z) = (\text{COMPOSE}(z, 1, Q, \sigma), \cdots, \text{COMPOSE}(z, N, Q, \sigma))$. This horizontal arity-wise compositions can be layered vertically, allowing the composition of predicates whose arities differ more than 1 (e.g., 2 layers of NLM can combine unary and quaternary predicates).

Two minor modification was made from the original paper. First, we use a slightly different tensor shapes: For the notational conveniency, we use hypercube-dimensional tensors of shape $\mathbb{B}^{O.^n.O \times P/n}$, instead of the original formulation $\mathbb{B}^{O \times O-1 \times \ldots O-n-1 \times P/n}$ which tries to reduce the size by disallowing the duplicates in the parameters. Our modification does not significantly affect the complexity of the representation because the original representation also has $O(O^n)$ complexity.

Second, we do not use the nullary predicates $z/0$ in order to disallow VAEs from encoding environment-specific information in it.

## A.3 LEARNING DISCRETE LATENT DYNAMICS USING BACK-TO-LOGIT

APPLY$(\boldsymbol{a}^i, \boldsymbol{z}^{i,0})$ is an arbitrary MLP, i.e., a neural black-box function that does not directly translates to a STRIPS action model, preventing efficient search with state-of-the-art classical planner. Cube-Space AE (Asai & Muise, 2020) addresses this issue by Back-To-Logit technique which replaces the MLP. Back-to-Logit places a so-called *cube-like graph prior* on the binary latent space / transitions. To understand the prior, the background of *cube-like graph* is necessary.

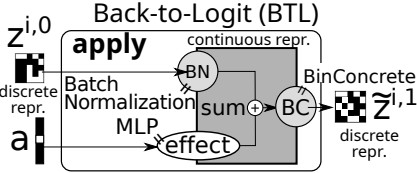

Figure 7: Back-To-Logit architecture (repost of Fig. 2).

### A.3.1 CUBE-LIKE GRAPHS AND ITS EQUIVALENCE TO STRIPS

Asai & Muise (2020) identified that state transition graphs of STRIPS planning problems form a graph class called directed *cube-like graph* (Payan, 1992). A cube-like graph is a simple[2] undirected graph $G(S, D) = (V, E)$ defined by sets $S$ and $D$. Each node $v \in V$ is a finite subset of $S$, i.e., $v \subseteq S$. The set $D$ is a family of subsets of $S$, and for every edge $e = (v, w) \in E$, the symmetric difference $d = v \oplus w = (v \setminus w) \cup (w \setminus v)$ must belong to $D$. For example, a unit cube is a cube-like graph because $S = \{x, y, z\}, V = \{\emptyset, \{x\}, \ldots \{x, y, z\}\}, E = \{(\emptyset, \{x\}), \ldots ((\{y, z\}, \{x, y, z\}))\}, D = \{\{x\}, \{y\}, \{z\}\}$. The set-based representation can be alternatively represented as a bit-vector, e.g., $V' = \{(0, 0, 0), (0, 0, 1), \ldots (1, 1, 1)\}$. We denote a one-to-one $|S|$-bit vector assignment as $f : V \to V' = \{0, 1\}^{|S|}$.

In STRIPS modeling, we use a directed version of this graph class. For every edge $e = (v, w) \in E$, there is a pair of sets $d = (d^+, d^-) = (w \setminus v, v \setminus w) \in D$ which satisfies the asymmetric difference $w = (v \setminus d^-) \cup d^+$. It is immediately obvious that this graph class corresponds to the relationship between binary states and action effects in STRIPS, $s' = (s \setminus \text{DEL}(()a)) \cup \text{ADD}(()a)$. As a special case, in an undirected STRIPS planning problem where all actions are reversible (i.e., for any action $a$ such that $t = a(s)$, there is an action $a^{-1}$ that is able to reverse the effect $s = a^{-1}(t)$), the state space graph is equivalent to an undirected cube-like graph.

### A.3.2 EDGE CHROMATIC NUMBER OF CUBE-LIKE GRAPHS AND THE NUMBER OF ACTION SCHEMA IN STRIPS

Edge coloring of cube-like graphs provides an intuitive understanding of the action schema expressible in STRIPS formalism. Consider coloring a graph which forms a unit cube (Fig. 8) and whose node embeddings correspond to the latent space of some images. A cube-like graph on the left can be efficiently colored (i.e., by fewer colors) by the difference between the neighboring embeddings. Edges can be categorized into 3 labels $(0, 0, \oplus 1)$, $(0, \oplus 1, 0)$ and $(\oplus 1, 0, 0)$ (6 labels if directed), where each label is assigned to 4 edges which share the same node embedding differences, as depicted by the upward arrows with the common node difference $(0, 0, \oplus 1)$ in the figure. The set of node embedding differences corresponds to the set $D$, and each element of $D$ represents an action, e.g., when the node embeddings are decoded into images, moving toward positive direction of $x$-axis may result in the "0" tile in the image moving to the right, and moving toward positive direction of $z$-axis may result in the "2" tile in the image moving to the right — effectively making actions compositional. In contrast, the graph on the right has node embeddings that are randomly shuffled. Despite having the same topology and the same embedding size, this graph lacks the common patterns in the embedding differences like we saw on the left, thus cannot be efficiently colored by the node differences.

---

[2]No duplicated edges between the same pair of nodes

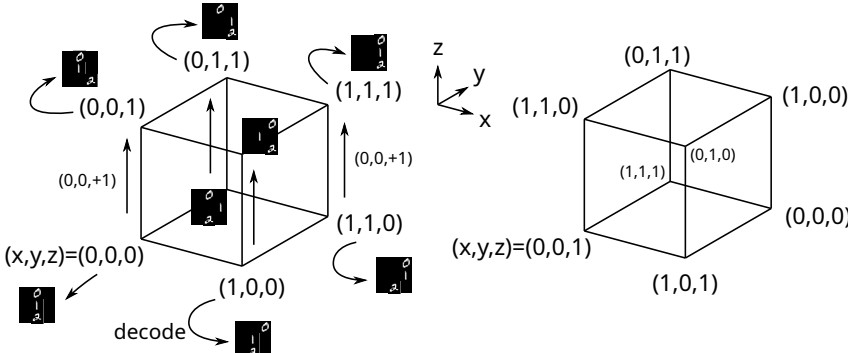

Figure 8: (Left) A graph representing a 3-dimensional cube which is a cube-like graph. (Right) A graph whose shape is identical to the left, but whose unique node embeddings are randomly shuffled.

As such, cube-like graphs can be characterized by the edge chromatic number (minimum edge coloring) according to the node differences. We now discuss some theoretical properties of the graph coloring with and without the assumptions on the colors and the node differences.

**Theorem 1** (Edge chromatic number Vizing (1965)). *Let the edge chromatic number $c(G)$ of an undirected graph $G$ be the number of colors in a minimum edge coloring. Then $c(G)$ is either $\Delta$ or $\Delta + 1$, where $\Delta$ is the maximum degree of the nodes.*

**Theorem 2.** *Mapping $E \to D$ provides an edge coloring and thus $c(G) \leq \min_f |D|$.*

*Proof.* $f$ is one-to-one: $w \neq w' \iff f(w) \neq f(w')$. For any set $X$, $f(w) \neq f(w') \iff X \oplus f(w) \neq X \oplus f(w')$. For any adjacent edges $(v, w)$ and $(v, w')$, $w \neq w'$ because $G$ is simple (at most one edge between any nodes), thus $f(v) \oplus f(w) \neq f(v) \oplus f(w')$. The remainder follows from the definition. □

For Thm. 2, equality holds for hypercubes. For a certain embedding dimension $|S|$, there are graph instances where $c(G) < \min_f |D|$ (Fig. 9, left). We "proved" this with an Answer Set Programming solver Potassco Gebser et al. (2011).

```
1 ● 4 ● 7 ● a ● d ●          (:action c1 :precondition (and) :effect
  |   |   |   |   |            (and (when (and (z0) (not (z1)) (z2))
2 ● 5 ● 8 ● b ● e ●               (and (z0) (z1) (not (z2)))))
  |   |   |   |   |            ...
3 ● 6 ● 9 ● c ● f ●          V1=(1,0,1) ●————color:c1————● V2=(1,1,0)
```

Figure 9: (left): An undirected graph consisting of 5 disconnected 2-star graphs It has $c(G) = 2$. When $|S| = 4$, no assignments satisfy $|D| = 2$. (An assignment with $|D| = 4$ exists.) (right): Single conditional effect can encode an arbitrary transition.

Based on these results, we next consider the minimum number of actions required to model an undirected cube-like graph $G(S, D)$ as a planning model. We restrict ourselves to a precondition-free planning domains in order to focus on the action effects. We first need a lemma in order to focus on undirected graphs.

**Lemma 1.** *Let $P$ be a precondition-free grounded STRIPS planning problem which contains irreversible actions. There is a corresponding planning problem $P'$ whose state transition graph is identical to that of $P$ and whose precondition relaxation $P''$ is reversible.*

*Proof.* For any irreversible action $a$ in $P$, add a set of actions $A^{-1}(a)$ whose size is $|A^{-1}(a)| = 2^{|\text{ADD}(()a) \cup \text{DEL}(()a)|}$. Each action $a' \in A^{-1}(a)$ contains those effects which encode one of the possible non-deterministic outcomes of reversing $a$, and contains an unsatisfiable precondition (e.g., adding a new proposition whose value is constantly false). Then the state transition graphs of $P$ and $P'$ are identical and $P''$ is reversible. □

Since we assume precondition-free domains, for any $P$ we could instead consider $P''$ to discuss the effects of reversible planning domains. Let $G$ be an undirected cube-like graph which is isomorphic to the state transition graph of a precondition-free reversible planning model $P$.

**Theorem 3.** *Let $P_c$ be another planning problem definition which models $G$. The action effects in $P_c$ are allowed to use conditional effects. Then the minimum number of actions in $P_c$ required to model $G$ is $c(G)$.*

*Proof.* For each color $c \in 1..c(G)$ and for each edge $(v, w)$ colored as $c$, and for each propositional value $f(w)_i \in \{0, 1\}$, we add a conditional effect to $c$-th action. If $f(w)_i = 0$, add a delete-effect, and $f(w)_i = 1$, add an add-effect. The effect is conditioned by the full conjunction of $f(v)$ using negative preconditions. See Fig. 9 (right), where all effects are put in one conditional effect. $\square$

**Theorem 4.** *The minimum number of actions required to model $G$ without conditional effects (i.e., by STRIPS effects) is $\min_f 2|D|$.*

*Proof.* Each $d \in D$ needs 2 actions for forward/backward directions. $\square$

Thm. 3 indicates that conditional effects can compact as many edges as possible into just $A = \Delta$ or $\Delta + 1$ actions regardless of the nature of the transitions, while Thm. 2 and Thm. 4 indicate that STRIPS effects require a larger number of actions. Therefore, merely assigning action symbols to state transitions (which is equivalent to edge coloring) does not result in a compact STRIPS model. Notice that the vanilla MLP AAE binds an unrestricted edge chromatic number $c(G) = \Delta$ or $\Delta + 1$ by the maximum number of action labels $A$ (a hyperparameter of the network), but does not bind $2|D|$, the edge chromatic number in terms of neighboring node embedding differences. In order to find a compact STRIPS action model, we should instead bind $2|D|$ by $A$ and restrict latent state transitions to follow a STRIPS transition rule.

In order to constrain the learned latent state space of the environment to a cube-like graph, we propose *Cube-Space AutoEncoder*. We first explain a vanilla *Space AutoEncoder*, an architecture that jointly learns the state and the action model by combining the SAE and the AAE into a single network. We then modify the APPLY progression to restrict state transitions. Due to the flexibility of neural networks, the loss enhanced by the restriction automatically propagates to the state representation, i.e., it modifies a state representation in order to reduce the loss produced by the restricted action model.

### A.3.3 VANILLA SPACE AUTOENCODER

The vanilla Space AutoEncoder (Fig. 10, right) connects the SAE and AAE subnetworks. The necessary modification, apart from connecting them, is the change in loss function. The original AAE was trained to optimize the distance between the binary vectors using binary crossentropy (BCE), which is assymmetric in definition: $\text{BCE}(x, y) = x \log y + (1 - x) \log(1 - y)$. While this was not problematic in the AAE which uses the fixed state representation $x$ and the successor prediction $y$, it is more natural for Vanilla Space AE to use a symmetric loss that equally affects $x$ and $y$.

In addition to the loss for the successor prediction in the latent space, we also ensure that the predicted successor state can be decoded back to the correct image $o^{i,1}$. Thus, the total loss is a sum of the losses for: (1) the main reconstructions $\ell(o^{i,0}, \tilde{o}^{i,0})$ and $\ell(o^{i,1}, \tilde{o}^{i,1})$, (2) the successor latent state reconstruction $\ell(z^{i,1}, \tilde{z}^{i,1})$, (3) the image reconstruction from the predicted successor $\ell(o^{i,1}, \tilde{\tilde{o}}^{i,1})$, and (4) the KL regularization. We call the second term as *direct loss*.

We next formally analyze this training objective. Given an observed transition $(o^0, o^1)$, we assume that $o^0$ follows $\mathcal{N}(\tilde{o}^0, \sigma_0)$ and $o^1$ follows $\mathcal{N}(\tilde{\tilde{o}}^0, \sigma_1)$ (See Sec. 2.2 for explanation). The maximization objective is a log-likelihood of observing a pair of states $o^0, o^1$.

We iteratively derive the lower bounds (ELBO) by inserting several latent variables. We first introduce $z^0, z^1$:

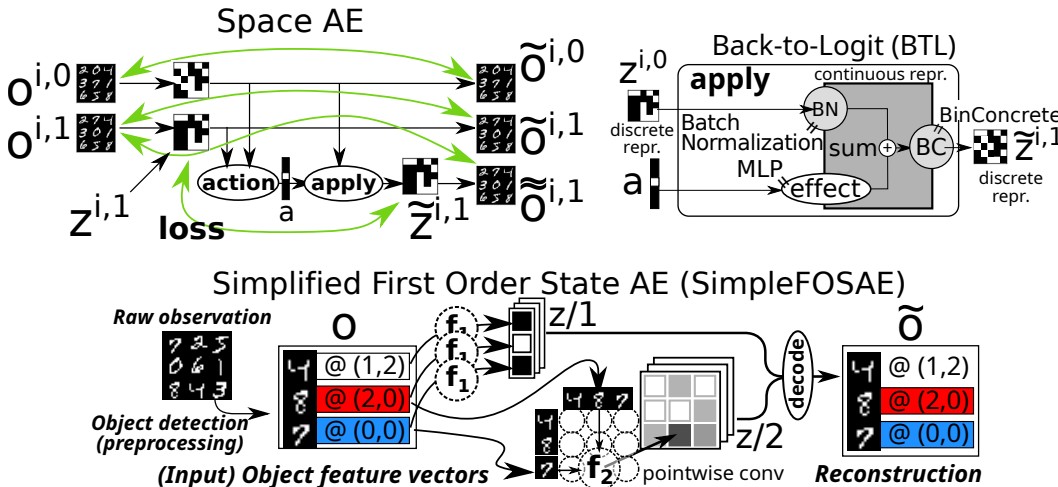

Figure 10: The illustration of State AutoEncoder, Action AutoEncoder, and the end-to-end combinations of the two.

$$\log p(\boldsymbol{o}^0, \boldsymbol{o}^1) \geq - D_{\mathrm{KL}}(q(\boldsymbol{z}^0, \boldsymbol{z}^1 \mid \boldsymbol{o}^0, \boldsymbol{o}^1) || p(\boldsymbol{z}^0, \boldsymbol{z}^1))$$
$$+ \mathbb{E}_{q(\boldsymbol{z}^0, \boldsymbol{z}^1 | \boldsymbol{o}^0, \boldsymbol{o}^1)}[\log p(\boldsymbol{o}^0, \boldsymbol{o}^1 \mid \boldsymbol{z}^0, \boldsymbol{z}^1)]. \tag{1}$$

The first term of Eq. 1 is the KL divergence for $\boldsymbol{z}^0, \boldsymbol{z}^1$. Since all latent variables are assumed to be independent (mean-field assumption), $p(\boldsymbol{z}^0, \boldsymbol{z}^1) = p(\boldsymbol{z}^0)p(\boldsymbol{z}^1)$, where $p(\boldsymbol{z}^0), p(\boldsymbol{z}^1)$ are respectively the prior distributions for the binary latent variable $\boldsymbol{z}^0$ and $\boldsymbol{z}^1$ that we discussed in Sec. **??**. $q(\boldsymbol{z}^0, \boldsymbol{z}^1 \mid \boldsymbol{o}^0, \boldsymbol{o}^1)$ can also be decomposed because, in the encoder modeled by $q$, $\boldsymbol{z}^0$ depends only on $\boldsymbol{o}^0$ and $\boldsymbol{z}^1$ depends only on $\boldsymbol{o}^1$. Therefore, the entire KL divergence is divided into individual KL divergence:

$$D_{\mathrm{KL}}(q(\boldsymbol{z}^0, \boldsymbol{z}^1 \mid \boldsymbol{o}^0, \boldsymbol{o}^1) || p(\boldsymbol{z}^0, \boldsymbol{z}^1)) = D_{\mathrm{KL}}(q(\boldsymbol{z}^0 \mid \boldsymbol{o}^0, \boldsymbol{o}^1) || p(\boldsymbol{z}^0)) + D_{\mathrm{KL}}(q(\boldsymbol{z}^1 \mid \boldsymbol{o}^0, \boldsymbol{o}^1) || p(\boldsymbol{z}^1))$$
$$= D_{\mathrm{KL}}(q(\boldsymbol{z}^0 \mid \boldsymbol{o}^0) || p(\boldsymbol{z}^0)) + D_{\mathrm{KL}}(q(\boldsymbol{z}^1 \mid \boldsymbol{o}^1) || p(\boldsymbol{z}^1)). \tag{2}$$

We next decompose the second term in Eq. 1 as shown in Eq. 3. This derivation is possible because $\tilde{\boldsymbol{o}}^0$ and $\tilde{\tilde{\boldsymbol{o}}}^1$ are generated independently in the decoder (the network that produces $\tilde{\boldsymbol{o}}^0, \tilde{\tilde{\boldsymbol{o}}}^1$ from $\boldsymbol{z}^0, \boldsymbol{z}^1$) modeled by $p$. The first term in Eq. 3 corresponds to a reconstruction loss for $\tilde{\boldsymbol{o}}^0$ (MSE due to $p(\boldsymbol{o}^0 \mid \boldsymbol{z}^0) = \mathcal{N}(\tilde{\boldsymbol{o}}^0, \sigma_0)$).

$$\log p(\boldsymbol{o}^0, \boldsymbol{o}^1 \mid \boldsymbol{z}^0, \boldsymbol{z}^1) = \log p(\boldsymbol{o}^0 \mid \boldsymbol{z}^0, \boldsymbol{z}^1) + \log p(\boldsymbol{o}^1 \mid \boldsymbol{z}^0, \boldsymbol{z}^1)$$
$$= \log p(\boldsymbol{o}^0 \mid \boldsymbol{z}^0) + \log p(\boldsymbol{o}^1 \mid \boldsymbol{z}^0, \boldsymbol{z}^1). \tag{3}$$

Next, we derive the lower bound of the second term in Eq. 3 by introducing a one-hot categorical latent variable $\boldsymbol{a}$ for action labels, and its prior distribution $p(\boldsymbol{a} \mid \boldsymbol{z}^0, \boldsymbol{z}^1) = \mathbf{Cat}(\mathbf{1}/A)$ (uniform categorical distribution of $A$ categories).

$$\log p(\boldsymbol{o}^1 \mid \boldsymbol{z}^0, \boldsymbol{z}^1) \geq - D_{\mathrm{KL}}(q(\boldsymbol{a} \mid \boldsymbol{o}^1, \boldsymbol{z}^0, \boldsymbol{z}^1) || p(\boldsymbol{a} \mid \boldsymbol{z}^0, \boldsymbol{z}^1))$$
$$+ \mathbb{E}_{q(\boldsymbol{a} | \boldsymbol{o}^1, \boldsymbol{z}^0, \boldsymbol{z}^1)}[\log p(\boldsymbol{o}^1 \mid \boldsymbol{a}, \boldsymbol{z}^0, \boldsymbol{z}^1)]. \tag{4}$$

Finally, to complete the Vanilla Space AE network, we should model the second term in Eq. 4 using another latent variable $\tilde{\boldsymbol{z}}^1$. We further derive the lower bound using the same reformulation:

$$\log p(\boldsymbol{o}^1 \mid \boldsymbol{a}, \boldsymbol{z}^0, \boldsymbol{z}^1) \geq - D_{\mathrm{KL}}(q(\tilde{\boldsymbol{z}}^1 \mid \boldsymbol{o}^1, \boldsymbol{a}, \boldsymbol{z}^0, \boldsymbol{z}^1) || p(\tilde{\boldsymbol{z}}^1 \mid \boldsymbol{a}, \boldsymbol{z}^0, \boldsymbol{z}^1))$$
$$+ \mathbb{E}_{q(\tilde{\boldsymbol{z}}^1 \mid \boldsymbol{o}^1, \boldsymbol{a}, \boldsymbol{z}^0, \boldsymbol{z}^1)}[\log p(\boldsymbol{o}^1 \mid \tilde{\boldsymbol{z}}^1, \boldsymbol{a}, \boldsymbol{z}^0, \boldsymbol{z}^1)]. \tag{5}$$

The second term of Eq. 5 is a reconstruction loss for $\tilde{\tilde{\boldsymbol{o}}}^1$ which can be computed by assuming $p(\boldsymbol{o}^1 \mid \tilde{\boldsymbol{z}}^1, \boldsymbol{a}, \boldsymbol{z}^0, \boldsymbol{z}^1) = \mathcal{N}(\tilde{\tilde{\boldsymbol{o}}}^1, \sigma_1)$. However, what should we use as a prior distribution $p(\tilde{\boldsymbol{z}}^1 \mid \boldsymbol{a}, \boldsymbol{z}^0, \boldsymbol{z}^1)$ in the first term (KL divergence)? We set it to be $q(\boldsymbol{z}^1 \mid \boldsymbol{o}^1)$, because the choice of a prior is arbitrary and we want that the distribution of the predicted successor state $\tilde{\boldsymbol{z}}^1$ to be identical to the distribution of the successor state $\boldsymbol{z}^1$ directly encoded from the input. As a result, the total maximization objective is derived as follows:

$$
\begin{array}{lr}
\log p(\boldsymbol{o}^0, \boldsymbol{o}^1) & \text{Interpretation} \\
\geq - D_{\mathrm{KL}}(q(\boldsymbol{z}^0 \mid \boldsymbol{o}^0) || p(\boldsymbol{z}^0)) & \text{KL divergence for } \boldsymbol{z}^0 \text{ in Eq. 2} \\
- D_{\mathrm{KL}}(q(\boldsymbol{z}^1 \mid \boldsymbol{o}^1) || p(\boldsymbol{z}^1)) & \text{KL divergence for } \boldsymbol{z}^1 \text{ in Eq. 2} \\
+ \log p(\boldsymbol{o}^0 \mid \boldsymbol{z}^0) & \text{Reconstruction loss } \ell(\boldsymbol{o}^0, \tilde{\boldsymbol{o}}^0) \text{ in Eq. 3} \\
- D_{\mathrm{KL}}(q(\boldsymbol{a} \mid \boldsymbol{o}^1, \boldsymbol{z}^0, \boldsymbol{z}^1) || p(\boldsymbol{a} \mid \boldsymbol{z}^0, \boldsymbol{z}^1)) & \text{KL divergence for } \boldsymbol{a} \text{ in Eq. 4} \\
- D_{\mathrm{KL}}(q(\tilde{\boldsymbol{z}}^1 \mid \boldsymbol{o}^1, \boldsymbol{a}, \boldsymbol{z}^0, \boldsymbol{z}^1) || q(\boldsymbol{z}^1 \mid \boldsymbol{o}^1)) & \text{KL divergence between } \boldsymbol{z}^1 \text{ and } \tilde{\boldsymbol{z}}^1 \text{ in Eq. 5} \\
& = \text{direct loss } \ell(\boldsymbol{z}^1, \tilde{\boldsymbol{z}}^1) \\
+ \log p(\boldsymbol{o}^1 \mid \tilde{\boldsymbol{z}}^1, \boldsymbol{a}, \boldsymbol{z}^0, \boldsymbol{z}^1). & \text{Reconstruction loss } \ell(\boldsymbol{o}^1, \tilde{\tilde{\boldsymbol{o}}}^1) \text{ in Eq. 5} \tag{6}
\end{array}
$$

Direct loss $\ell(\boldsymbol{z}^1, \tilde{\boldsymbol{z}}^1)$ can be computed by the same method introduced in (Sec. **??**). We assume $q(\tilde{\boldsymbol{z}}^1 \mid \boldsymbol{o}^1, \boldsymbol{a}, \boldsymbol{z}^0, \boldsymbol{z}^1) = \mathrm{Bernoulli}(\tilde{\boldsymbol{q}}^1)$ and $q(\boldsymbol{z}^1 \mid \boldsymbol{o}^1) = \mathrm{Bernoulli}(\boldsymbol{q}^1)$ where $\tilde{\boldsymbol{q}}^1 = \sigma(\tilde{\boldsymbol{l}}^1)$, $\boldsymbol{q}^1 = \sigma(\boldsymbol{l}^1)$, and $\tilde{\boldsymbol{l}}^1, \boldsymbol{l}^1$ are the inputs to the corresponding Binary Concrete activation. Then the KL divergence is computed as

$$D_{\mathrm{KL}}(q(\tilde{\boldsymbol{z}}^1 \mid \boldsymbol{o}^1, \boldsymbol{a}, \boldsymbol{z}^0, \boldsymbol{z}^1) || q(\boldsymbol{z}^1 \mid \boldsymbol{o}^1)) = D_{\mathrm{KL}}(\mathrm{Bernoulli}(\tilde{\boldsymbol{q}}^1) || \mathrm{Bernoulli}(\boldsymbol{q}^1))$$
$$= \tilde{\boldsymbol{q}}^1 \log \frac{\tilde{\boldsymbol{q}}^1}{\boldsymbol{q}^1} + (1 - \tilde{\boldsymbol{q}}^1) \log \frac{1 - \tilde{\boldsymbol{q}}^1}{1 - \boldsymbol{q}^1}. \tag{7}$$

Coincidentally, this is $D_{\mathrm{KL}}(q(\tilde{\boldsymbol{z}}^1 \mid \boldsymbol{o}^1, \boldsymbol{z}^0, \boldsymbol{a}) || \mathrm{Bernoulli}(0.5)) + \log 2 + \mathrm{BCE}(\tilde{\boldsymbol{q}}^1, \boldsymbol{q}^1)$, whose last binary cross entropy term similar to the loss function of AAE.

**Implementation note:** If we instead optimize $D_{\mathrm{KL}}(q(\tilde{\boldsymbol{z}}^1 \mid \boldsymbol{o}^1, \boldsymbol{z}^0, \boldsymbol{a}) || \mathrm{Bernoulli}(0.5)) + \log 2 + \mathrm{BCE}(\tilde{\boldsymbol{z}}^1, \boldsymbol{z}^1)$, the training becomes slower due to the unnecessary logistic noise inside BC of $\tilde{\boldsymbol{z}}^1 = \mathrm{BC}(\tilde{\boldsymbol{l}}^1)$ and $\boldsymbol{z}^1 = \mathrm{BC}(\boldsymbol{l}^1)$.

**Implementation note:** It is important to bootstrap various terms in the optimization objective. In our experience, delaying the application of the direct loss until a certain epoch seems crucial, and we suspect the reason is due to the relatively short network distance between the two latent spaces $\boldsymbol{z}^{i,1}/\tilde{\boldsymbol{z}}^{i,1}$ compared to $\boldsymbol{o}^{i,1}/\tilde{\tilde{\boldsymbol{o}}}^{i,1}$. If we enable the direct loss from the beginning, the total loss does not converge because $\tilde{\boldsymbol{z}}^{i,1}$ prematurely converges to $\boldsymbol{z}^{i,1}$ causing a mode-collapse (e.g. all 0), before the image en/decoder learns a meaningful latent representation.

### A.3.4 CUBE-SPACE AE

Cube-Space AE modifies the APPLY network so that it directly predicts the effects without taking the current state as the input, and logically computes the successor state based on the predicted effect and the current state.

A naive implementation of such a network is shown in Fig. 11 (left). The EFFECT network predicts a binary tensor of $F \times 3$ using $F$-way Gumbel-Softmax of 3 categories. Each Gumbel Softmax corresponds to one bit in the $F$-bit latent space and 3 classes correspond to the add effect, delete effect and NOP, only one of which is selected by the one-hot vector. The effects are applied to the

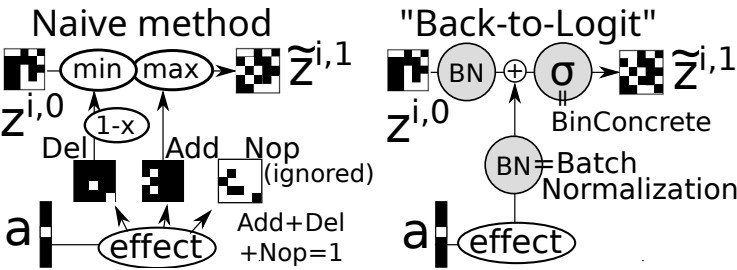

Figure 11: A naive and a Back-to-Logit implementation of the APPLY module of the Cube-Space AE.

current state either by a max/min operation or its smooth variants (smooth min/max). Formally, the naive Cube-Space AE is formulated as follows:

$$\mathbb{B}^F \ni \widetilde{\boldsymbol{z}}^{i,1} = \text{APPLY}(\boldsymbol{z}^{i,0}, \boldsymbol{a}^i) = \min(\max(\boldsymbol{z}^{i,0}, \text{ADD}(()\boldsymbol{a}^i)), 1 - \text{DEL}(()\boldsymbol{a}^i)) \quad \text{or} \tag{8}$$

$$\text{smin}(\text{smax}(\boldsymbol{z}^{i,0}, \text{ADD}(()\boldsymbol{a}^i)), 1 - \text{DEL}(()\boldsymbol{a}^i)), \quad \text{where} \tag{9}$$

$$\text{EFFECT}(()\boldsymbol{a}^i) = \text{GS}(\text{MLP}(\boldsymbol{a}^i)) \in \mathbb{B}^{F \times 3}, \qquad \text{ADD}(()\boldsymbol{a}^i) = \text{EFFECT}(()\boldsymbol{a}^i)_0,$$

$$\text{DEL}(()\boldsymbol{a}^i) = \text{EFFECT}(()\boldsymbol{a}^i)_1, \qquad \text{NOP}(\boldsymbol{a}^i) = \text{EFFECT}(()\boldsymbol{a}^i)_2,$$

$$\text{smax}(x, y) = \log(e^x + e^y), \qquad \text{smin}(x, y) = -\log(e^{-x} + e^{-y}). \tag{10}$$

While being intuitive, we found these naive implementations extremely difficult to train. Our contribution to the architecture is *Back-to-Logit* (BtL, Fig. 11, right), *a generic approach that computes a logical operation in the continuous logit space*. We re-encode a logical, binary vector back to a continuous, logit representation by an monotonic function $m$. This monotonicity preserves the order between true (1) and false (0) even after transformed into real numbers. We then apply effects by adding the continuous state vector with a continuous *effect* vector. The effect vector is produced by applying an MLP named EFFECT to the action vector $\boldsymbol{a}^i$. After adding the continuous vectors, we re-activate the result with a discrete activation (Binary Concrete). Formally,

$$\widetilde{\boldsymbol{z}}^{i,1} = \text{APPLY}(\boldsymbol{z}^{i,0}, \boldsymbol{a}^i) = \text{BC}(m(\boldsymbol{z}^{i,0}) + \text{EFFECT}(()\boldsymbol{a}^i)). \tag{11}$$

We found that an easy and successful way to implement $m$ is *Batch Normalization* Ioffe & Szegedy (2015), a method that was originally developed for addressing the covariate shift in the deep neural networks.

**Additional background:** For simplicity, we consider a scalar operation, which can be applied to vectors element-wise. During the batch training of the neural network, Batch Normalization layer BN($x$) takes a minibatch input $B = \{x^1 \ldots x^{|B|}\}$, computes the mean $\mu$ and variance $\sigma^2$, then shift and scale each $x^i$ so that the result has a mean of 0 and a variance of 1. It then shifts and scales the results by two trainable coefficients $\gamma$ and $\beta$. Formally,

$$\forall x^i \in B; \ \text{BN}(x^i) = \frac{x^i - \mu}{\sigma^2}\gamma + \beta. \tag{12}$$

While rescaling the normalized result by $\gamma$ and $\beta$ seem to negate the original purpose of normalization, the presense of normalization to mean 0 / variance 1 is crucial. Notice that the first scaling depends on other training examples in the same minibatch, while $\gamma$ and $\beta$ is not dynamically adjusted for the minibatch. For example, imagine two minibatches $B_1$ and $B_2$, where $B_1$ accidentally tends to contain larger values than $B_2$ but the variance within $B_1$ and $B_2$ are the same. The bias is canceled by the normalization, and thus the outputs rescaled by $\gamma$ and $\beta$ are computed based on a relative scale inside the corresponding minibatch.

**Implementation note:** Since $\boldsymbol{a}^i$ is a probability vector over $A$ action ids and $\boldsymbol{a}^i$ eventually converges to a one-hot vector due to Gumbel-Softmax annealing, the additional MLP can be merely a linear

embedding, i.e., $\text{EFFECT}(()\boldsymbol{a}^i) = \boldsymbol{E}\boldsymbol{a}^i$, where $\boldsymbol{E} \in \mathbb{R}^{F \times A}$ and $\boldsymbol{z} \in \mathbb{B}^F$. It also helps the training if we apply BatchNorm on the effect vector. Therefore, a recommended implementation is

$$\text{APPLY}(a, \boldsymbol{z}^{i,0}) = \text{BC}(\text{BN}(\boldsymbol{z}^{i,0}) + \text{BN}(\boldsymbol{E}\boldsymbol{a}^i)) \tag{13}$$

where $\text{EFFECT}(()\boldsymbol{a}^i) = \text{BN}(\boldsymbol{E}\boldsymbol{a}^i)$.

### A.3.5 BACK-TO-LOGIT AND ITS EQUIVALENCE TO STRIPS

States learned by BTL has the following property:

**Theorem 5.** *(Asai & Muise, 2020) Under the same action $a$, state transitions are bitwise monotonic, deterministic, and restricted to three mutually exclusive modes, i.e., for each bit $j$:*

$$(add:) \; \forall i; (\boldsymbol{z}_j^{i,0}, \boldsymbol{z}_j^{i,1}) \in \{(0,1),(1,1)\} \; i.e. \; \boldsymbol{z}_j^{i,0} \leq \boldsymbol{z}_j^{i,1} \tag{14}$$

$$(del:) \; \forall i; (\boldsymbol{z}_j^{i,0}, \boldsymbol{z}_j^{i,1}) \in \{(1,0),(0,0)\} \; i.e. \; \boldsymbol{z}_j^{i,0} \geq \boldsymbol{z}_j^{i,1} \tag{15}$$

$$(nop:) \; \forall i; (\boldsymbol{z}_j^{i,0}, \boldsymbol{z}_j^{i,1}) \in \{(0,0),(1,1)\} \; i.e. \; \boldsymbol{z}_j^{i,0} = \boldsymbol{z}_j^{i,1} \tag{16}$$

This theorem guarantees that each action deterministically sets a certain bit on and off in the binary latent space. Therefore, the actions and the transitions satisfy the STRIPS state transition rule $s' = (s \setminus \text{DEL}(()a)) \cup \text{ADD}(()a)$, thus enabling a direct translation from neural network weights to PDDL modeling language.

The proof is straightforward from the monotonicity of the BatchNorm and Binary Concrete. Note that we assume BatchNorm's additional scale parameter $\gamma$ is kept positive or disabled.

*Proof.* For readability, we omit $j$ and assumes a 1-dimensional case. Let $e = \text{EFFECT}(()\boldsymbol{a}) \in \mathbb{R}$, which is a constant for the fixed input $\boldsymbol{a}$. At the limit of annealing, Binary Concrete BC becomes a STEP function, which is also monotonic. BN is monotonic because we assumed the scale parameter $\gamma$ of BN is positive, and the main feature of BN also only scales the variance of the batch, which is always positive. Then we have

$$\boldsymbol{z}^{i,1} = \text{STEP}(\text{BN}(\boldsymbol{z}^{i,0}) + e). \tag{17}$$

The possible values a pair $(\boldsymbol{z}^{i,0}, \boldsymbol{z}^{i,1})$ can have is $(0,0),(0,1),(1,0),(1,1)$. Since both STEP and BN are deterministic at the testing time (See Ioffe & Szegedy (2015)), we consider the deterministic mapping from $\boldsymbol{z}^{i,0}$ to $\boldsymbol{z}^{i,1}$. There are only 4 deterministic mappings from $\{0,1\}$ to $\{0,1\}$: $\{(0,1),(1,1)\}, \{(1,0),(0,0)\}, \{(0,0),(1,1)\}, \{(0,1),(1,0)\}$. Thus our goal is to show that the last mapping $\{(0,1),(1,0)\}$ is impossible in the latent space $\{\ldots(\boldsymbol{z}^{i,0}, \boldsymbol{z}^{i,1})\ldots\}$ produced by BTL.

To prove this, first, assume $(\boldsymbol{z}^{i,0}, \boldsymbol{z}^{i,1}) = (0,1)$ for some index $i$. Then

$$1 = \text{STEP}(\text{BN}(0) + e) \Rightarrow \text{BN}(0) + e > 0 \Rightarrow \text{BN}(1) + e > 0 \Rightarrow \forall i; \text{BN}(\boldsymbol{z}^{i,0}) + e > 0. \tag{18}$$

The second step is due to the monotonicity $\text{BN}(0) < \text{BN}(1)$. This shows $\boldsymbol{z}^{i,1}$ is constantly 1 regardless of $\boldsymbol{z}^{i,0}$, therefore it proves that $(\boldsymbol{z}^{i,0}, \boldsymbol{z}^{i,1}) = (1,0)$ cannot happen in any $i$.

Likewise, if $(\boldsymbol{z}^{i,0}, \boldsymbol{z}^{i,1}) = (1,0)$ for some index $i$,

$$0 = \text{STEP}(\text{BN}(1) + e) \Rightarrow \text{BN}(1) + e < 0 \Rightarrow \text{BN}(0) + e < 0 \Rightarrow \forall i; \text{BN}(\boldsymbol{z}^{i,0}) + e < 0. \tag{19}$$

Therefore, $\boldsymbol{z}^{i,1} = 0$ regardless of $\boldsymbol{z}^{i,0}$, and thus $(\boldsymbol{z}^{i,0}, \boldsymbol{z}^{i,1}) = (0,1)$ cannot happen in any $i$.

Finally, if the data points do not contain $(0,1)$ or $(1,0)$, then by assumption they do not coexist. Therefore, the embedding learned by BTL cannot contain $(0,1)$ and $(1,0)$ at the same time. $\square$

### A.4 EFFECT RULE EXTRACTION

To extract the effects of an action $\boldsymbol{a}$ from Cube-Space AE, we compute $\text{ADD}(\boldsymbol{a}) = \text{APPLY}(\boldsymbol{a}, \boldsymbol{0})$ and $\text{DEL}(\boldsymbol{a}) = 1 - \text{APPLY}(\boldsymbol{a}, \boldsymbol{1})$ for each action $\boldsymbol{a}$, where $\boldsymbol{0}, \boldsymbol{1}$ are vectors filled by zeros/ones and has the same size as the binary embedding. Since APPLY deterministically sets values to 0 or 1, feeding

these vectors is sufficient to see which bit it turns on and off. For each $j$-th bit that is 1 in each result, a corresponding proposition is added to the add/delete-effect, respectively.

In FOSAE++, we extracts the effects from parameter-bounded subspace $z_{\dagger}^{i,0}, z_{\dagger}^{i,1}$. The representation is a tuple $z_{\dagger} = (z_{\dagger}/1 \cdots z_{\dagger}/N)$, where $z_{\dagger}/n \in \mathbb{B}^{\#a \times \overset{n}{\ldots} \times \#a \times P/n}$. BTL then operates on flattened and concatenated binary vectors of size $\sum_n \#a^n P/n$: The input, the output, and the effect share this shape. We extract the effects from this BTL vector in the same manner as noted above. After the extraction, however, each bit is converted to a lifted predicate according to the position. For example, when a bit that corresponds to $z_{\dagger}/2_{1,2,5}$ has turned from 0 to 1, then the add-effect contains p5(?arg1, ?arg2), where ?arg1 is a parameter used in the lifted PDDL encoding.

We show an example of such a learned PDDL model below, obtained from 8-Puzzle with $P/1 = P/2 = 333$ (reformatted for readability). Note that we disabled the nullary predicates $P/0$ and $z/0$, which consumes the first 333 dimensions in the flattened vector. Another note is that we also count the number of appearances of each action in the training dataset. If an action label is never used in the dataset, it is not exported in the resulting PDDL output. Thus, the index for the action starts from 7 in the example.

```
(define (domain latent)
 (:requirements :strips :negative-preconditions)
 (:predicates
  (p333 ?x0)      ...   (p665 ?x0)
  (p666 ?x0 ?x1)  ...   (p998 ?x0 ?x1))

 (:action a7 :parameters (?x0) :precondition
  (and (p339 ?x0) (p388 ?x0) (p391 ?x0) (p398 ?x0) (p402 ?x0)
       (p420 ?x0) (p421 ?x0) (p446 ?x0) (p447 ?x0) (p473 ?x0)
       (p475 ?x0) (p489 ?x0) (p491 ?x0) (p502 ?x0) (p516 ?x0)
       (p559 ?x0) (p588 ?x0) (p615 ?x0) (p641 ?x0) (p648 ?x0)
       (p831 ?x0 ?x0) (p950 ?x0 ?x0) (not (p333 ?x0))
       (not (p371 ?x0)) (not (p375 ?x0)) (not (p388 ?x0))
       (not (p402 ?x0)) (not (p406 ?x0)) (not (p421 ?x0))
       (not (p447 ?x0)) (not (p454 ?x0)) (not (p504 ?x0))
       (not (p508 ?x0)) (not (p519 ?x0)) (not (p524 ?x0))
       (not (p526 ?x0)) (not (p562 ?x0)) (not (p584 ?x0))
       (not (p593 ?x0)) (not (p617 ?x0)) (not (p640 ?x0))
       (not (p652 ?x0)) (not (p824 ?x0 ?x0)) (not (p892 ?x0 ?x0))
       (not (p926 ?x0 ?x0)) (not (p975 ?x0 ?x0))
       (not (p994 ?x0 ?x0)))
  :effect
  (and (p349 ?x0) (p361 ?x0) (p366 ?x0) (p370 ?x0) (p371 ?x0)
       (p378 ?x0) (p381 ?x0) (p385 ?x0) (p388 ?x0) (p401 ?x0)
       (p408 ?x0) (p421 ?x0) (p432 ?x0) (p445 ?x0) (p454 ?x0)
       (p475 ?x0) (p491 ?x0) (p496 ?x0) (p502 ?x0) (p503 ?x0)
       (p504 ?x0) (p507 ?x0) (p517 ?x0) (p526 ?x0) (p550 ?x0)
       (p562 ?x0) (p563 ?x0) (p575 ?x0) (p584 ?x0) (p588 ?x0)
       (p599 ?x0) (p601 ?x0) (p607 ?x0) (p612 ?x0) (p617 ?x0)
       (p631 ?x0) (p640 ?x0) (p641 ?x0) (p647 ?x0) (p656 ?x0)
       (p663 ?x0) (p724 ?x0 ?x0) (p768 ?x0 ?x0) (p831 ?x0 ?x0)
       (p902 ?x0 ?x0) (p911 ?x0 ?x0) (p993 ?x0 ?x0) (not (p339 ?x0))
       (not (p355 ?x0)) (not (p365 ?x0)) (not (p391 ?x0))
       (not (p397 ?x0)) (not (p398 ?x0)) (not (p402 ?x0))
       (not (p406 ?x0)) (not (p422 ?x0)) (not (p446 ?x0))
       (not (p447 ?x0)) (not (p448 ?x0)) (not (p451 ?x0))
       (not (p456 ?x0)) (not (p472 ?x0)) (not (p473 ?x0))
       (not (p478 ?x0)) (not (p489 ?x0)) (not (p490 ?x0))
       (not (p495 ?x0)) (not (p516 ?x0)) (not (p518 ?x0))
       (not (p524 ?x0)) (not (p525 ?x0)) (not (p527 ?x0))
       (not (p534 ?x0)) (not (p544 ?x0)) (not (p559 ?x0))
```

```
        (not (p561 ?x0)) (not (p615 ?x0)) (not (p624 ?x0))
        (not (p629 ?x0)) (not (p642 ?x0)) (not (p646 ?x0))
        (not (p651 ?x0)) (not (p653 ?x0)) (not (p720 ?x0 ?x0))
        (not (p813 ?x0 ?x0)) (not (p824 ?x0 ?x0)) (not (p892 ?x0 ?x0))
        (not (p894 ?x0 ?x0)) (not (p926 ?x0 ?x0)) (not (p931 ?x0 ?x0))
        (not (p975 ?x0 ?x0)) (not (p994 ?x0 ?x0))))
 (:action a22 :parameters (?x0) :precondition
  ...
```

## A.5 PRECONDITION LEARNING WITH DYNAMICS REVERSED IN TIME

In the main text, we simplified the model by showing only the forward dynamics, i.e., the dynamics in the same direction as the time. This forward dynamics can model the effects (add/delete) of the actions. However, the forward dynamics is insufficient for learning the preconditions of the actions. The original CSAE paper (Asai & Muise, 2020) used an add-hoc method that extracts common bits of the current states.

In contrast, we added a network that uses the same BTL mechanism that is applied backward in time, i.e., predict the current state $z^{i,0}$ from a successor state $z^{i,1}$ and a one-hot action vector $a^i$. We named the network REGRESS($z^{i,1}, a^i$), alluding to the *regression planning* (Alcázar et al., 2013) literature.

In REGRESS, add-effects and delete-effects now correspond to *positive preconditions* and *negative preconditions*. A positive precondition (normal precondition) requires that a proposition is $\top$ prior to using an action. In contrast, a negative precondition requires that a proposition is $\bot$ prior to using an action. While negative preconditions are (strictly speaking) out of STRIPS formalism, it is commonly supported by the modern classical planners that participated in the recent competitions. To extract the preconditions from the network, we can use the same method used for extracting the effects from the progressive/forward dynamics.

The entire model thus looks as follows.

$$
\begin{aligned}
\text{(encoder)} \quad & z^{i,0}, z^{i,1} = \text{ENCODE}(o^{i,0}), \text{ENCODE}(o^{i,1}) \\
\text{(action parameters, uses NLMs)} \quad & x^i = \text{PARAMS}(z^{i,0}, z^{i,1}) \\
\text{(parameter-bound subspace extraction)} \quad & z^{i,0}_\dagger, z^{i,1}_\dagger = \text{BIND}(z^{i,0}, x^i), \text{BIND}(z^{i,1}, x^i) \\
\text{(action assignment)} \quad & a^i = \text{ACTION}(z^{i,0}_\dagger, z^{i,1}_\dagger) \\
\text{(bounded forward dynamics)} \quad & \tilde{z}^{i,1}_\dagger = \text{APPLY}(z^{i,0}_\dagger, a^i) \\
\text{(bounded backward dynamics)} \quad & \tilde{z}^{i,0}_\dagger = \text{REGRESS}(z^{i,1}_\dagger, a^i) \\
\text{(reflection to global forward dynamics)} \quad & \tilde{z}^{i,1} = z^{i,0} - \text{UNBIND}(z^{i,0}_\dagger, x^i) + \text{UNBIND}(\tilde{z}^{i,1}_\dagger, x^i) \\
\text{(reflection to global backward dynamics)} \quad & \tilde{z}^{i,0} = z^{i,1} - \text{UNBIND}(z^{i,1}_\dagger, x^i) + \text{UNBIND}(\tilde{z}^{i,0}_\dagger, x^i) \\
\text{(reconstructions)} \quad & \tilde{o}^{i,0}, \tilde{o}^{i,1} = \text{DECODE}(z^{i,0}), \text{DECODE}(z^{i,1}) \\
\text{(reconstruction based on forward dynamics)} \quad & \tilde{\tilde{o}}^{i,1} = \text{DECODE}(\tilde{z}^{i,1}) \\
\text{(reconstruction based on backward dynamics)} \quad & \tilde{\tilde{o}}^{i,0} = \text{DECODE}(\tilde{z}^{i,0})
\end{aligned}
$$

The total loss is $\ell(o^{i,0}, \tilde{o}^{i,0}) + \ell(o^{i,1}, \tilde{o}^{i,1}) + \ell(o^{i,1}, \tilde{\tilde{o}}^{i,1}) + \ell(o^{i,0}, \tilde{\tilde{o}}^{i,0}) + \ell(z^{i,1}, \tilde{z}^{i,1}) + \ell(z^{i,1}_\dagger, \tilde{z}^{i,1}_\dagger) + \ell(z^{i,0}, \tilde{z}^{i,0}) + \ell(z^{i,0}_\dagger, \tilde{z}^{i,0}_\dagger) + \text{Reg}.$

## B    IMPLEMENTATION DETAIL

We based our code on a publicly-available Latplan source code repository (https://github.com/guicho271828/latplan/), which is based on Keras Deep Learning library. The repository hosts its own Genetic Algorithm based hyperparameter tuner which we mention several times later.

### B.1    INPUT DATA FORMAT AND LOSS FUNCTIONS

As we discussed in the earlier appendix section, our final loss function consists of 9 terms: $\ell(\boldsymbol{o}^{i,0}, \tilde{\boldsymbol{o}}^{i,0}) + \ell(\boldsymbol{o}^{i,1}, \tilde{\boldsymbol{o}}^{i,1}) + \ell(\boldsymbol{o}^{i,1}, \tilde{\tilde{\boldsymbol{o}}}^{i,1}) + \ell(\boldsymbol{o}^{i,0}, \tilde{\tilde{\boldsymbol{o}}}^{i,0}) + \ell(\boldsymbol{z}^{i,1}, \tilde{\boldsymbol{z}}^{i,1}) + \ell(\boldsymbol{z}_{\dagger}^{i,1}, \tilde{\boldsymbol{z}}_{\dagger}^{i,1}) + \ell(\boldsymbol{z}^{i,0}, \tilde{\boldsymbol{z}}^{i,0}) + \ell(\boldsymbol{z}_{\dagger}^{i,0}, \tilde{\boldsymbol{z}}_{\dagger}^{i,0}) + \mathrm{Reg}$. We first describe the input data format that is shared among different domains, then the loss functions defined on it.

Our input/output format $\boldsymbol{o} \in \mathbb{R}^{O \times F}$ consists of $O$ objects (environment-dependent) each having $F$ features. $F$ features consists of image-based features and the coordinate / dimension-based features (Fig. 12). All image patches extracted from the observations are resized into a fixed height $H$, width $W$ and color channel $C = 3$. Each flattened object vector has the size $F = H \times W \times C + 4$. The last 4 dimensions contain the center coordinate and the actual height/width before the resizing. Out of the 9 terms in the total loss, the terms that apply to the object vectors of this form are $\ell(\boldsymbol{o}^{i,0}, \tilde{\boldsymbol{o}}^{i,0})$, $\ell(\boldsymbol{o}^{i,1}, \tilde{\boldsymbol{o}}^{i,1})$, $\ell(\boldsymbol{o}^{i,1}, \tilde{\tilde{\boldsymbol{o}}}^{i,1})$, $\ell(\boldsymbol{o}^{i,0}, \tilde{\tilde{\boldsymbol{o}}}^{i,0})$.

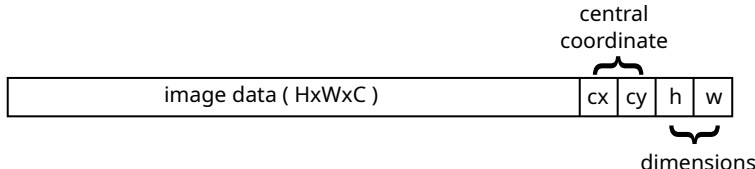

Figure 12: Data array representing a single object.

For this data format, the loss function consists of the mean square value of the image part, and the square sum of the coordinate / dimension parts. We do not average the losses for the coordinates/dimensions to avoid making the gradient minuscule. To further enhance this direction, we additionally have the *coordinate loss amplifier* $\lambda$ tuned by GA, as it is often the case that the object location has more visual impact on the reconstruction. Note that, for the tuning with the validation set and the evaluation with the test set, we set $\lambda = 1$ in order to obtain the consistent, comparable measuring. $\lambda$ is altered only during the training. Formally, for the $i$-th objects in the input $\boldsymbol{o}$ and the reconstruction $\tilde{\boldsymbol{o}}$, we define the loss as follows. These losses are averaged over the objects and the batch dimensions.

$$\ell(\boldsymbol{o}_i, \tilde{\boldsymbol{o}}_i) = \frac{1}{HWC} ||\boldsymbol{o}_{i,1..HWC} - \tilde{\boldsymbol{o}}_{i,1..HWC}||_2^2 + \lambda ||\boldsymbol{o}_{i,HWC..F} - \tilde{\boldsymbol{o}}_{i,HWC..F}||_2^2$$

We call the remaining losses except the regularization terms as "latent dynamics loss". They operate on the binary latent data activated by BinaryConcrete: $\ell(\boldsymbol{z}^{i,1}, \tilde{\boldsymbol{z}}^{i,1})$, $\ell(\boldsymbol{z}_{\dagger}^{i,1}, \tilde{\boldsymbol{z}}_{\dagger}^{i,1})$, $\ell(\boldsymbol{z}^{i,0}, \tilde{\boldsymbol{z}}^{i,0})$, $\ell(\boldsymbol{z}_{\dagger}^{i,0}, \tilde{\boldsymbol{z}}_{\dagger}^{i,0})$. However, note that during the training, all values used for the loss calculation are still continuous due to BinaryConcrete's annealing. This means we can't use Binary Cross Entropy BCE, the standard loss function for binary classifications, because the "training data" is also a noisy probability value. It is also in fact symmetric — as we discussed in the previous appendix sections, the role of these losses is not only just to obtain the accurate dynamics, but *also to shape the state representation toward a cube-like graph*. While there are several candidate loss functions that can be considered, we adapted Symmetric Cross Entropy (Wang et al., 2019) designed for noisy labels, which simply applies BCE in both ways. Formally, given $\mathrm{BCE}(\boldsymbol{z}, \tilde{\boldsymbol{z}}) = -\sum_i (\boldsymbol{z}_i \log \tilde{\boldsymbol{z}}_i + (1 - \boldsymbol{z}_i) \log(1 - \tilde{\boldsymbol{z}}_i))$,

$$\ell(\boldsymbol{z}, \tilde{\boldsymbol{z}}) = \mathrm{BCE}(\boldsymbol{z}, \tilde{\boldsymbol{z}}) + \mathrm{BCE}(\tilde{\boldsymbol{z}}, \boldsymbol{z}).$$

The remaining regularization losses include KL divergence for Discrete VAEs, as well as the L1 regularization for the latent vectors $\boldsymbol{z}^{i,0}, \boldsymbol{z}^{i,1}$ which were proven useful in Asai & Kajino (2019).

Finally, we define the magnitude and the warmup of each loss. The magnitude multiplies each loss and is tuned by the GA tuner. Similar to $\lambda$, the values are set to 1 during the evaluation. We have $\alpha$ for the L1 regularization, $\beta$ for KL divergence, and $\gamma$ for the latent dynamics loss.

The warmup mechanism works by setting these values to 0 until the training reaches a certain epoch defined by the ratio $r$ relative to the total number of epochs. We used the warmup $r_\alpha$, $r_\gamma$ for $\alpha$, $\gamma$, as well as $r_{\text{rec}}$ and the $r_{\text{dyn}}$ for the main reconstruction loss $\ell(\boldsymbol{o}^{i,0}, \tilde{\boldsymbol{o}}^{i,0}) + \ell(\boldsymbol{o}^{i,1}, \tilde{\boldsymbol{o}}^{i,1})$ and the dynamics-based reconstruction loss $\ell(\boldsymbol{o}^{i,1}, \tilde{\tilde{\boldsymbol{o}}}^{i,1}) + \ell(\boldsymbol{o}^{i,0}, \tilde{\tilde{\boldsymbol{o}}}^{i,0})$.

The motivation behind these magnitudes and warmups is to balance the speed of convergence of the various parts of the networks. Depending on the hyperparameters (depth, width of the layers), occasionally the network completely ignores the dynamics, falling into something similar to (but the mechanism will be very different from) a mode collapse where the effect caused by the dynamics is empty, e.g., the forward dynamics produces the same state as the current state.

Another failure mode of the network is that the dynamics loss is too strong, due to its BCE which could become too large compared to the reconstruction loss. As a result, the network learns a perfect but meaningless latent space dynamics which does not produce correct reconstructions. Tuning the warmup and balancing the losses addressed these issues.

## B.2 Network Detail

FOSAE++ consists of 6 networks: ENCODE, PARAMS, ACTION, APPLY, REGRESS, DECODE. Note that BIND and UNBIND are weight-less operations.

### B.2.1 Encoder

ENCODE can be divided into trivial continuous feature extraction phase (pre-encoder) and the actual FOSAE encoder. The feature extractor is a simple 1D pointwise convolution over the objects, i.e., same as applying the same Dense/Fully-Connected(FC) layer on each individual object. Its depth and the hidden layer width is tuned by the GA tuner. All activations except the last Binary Concrete are Rectified Linear Unit (Fukushima, 1980; Nair & Hinton, 2010). The architecture is illustrated in Fig. 12.

We should note that we assign the same number of predicates to different arities: $P/1 = P/2 \ldots = P/N$. In this Network Detail section, we denote $P = P/1 \ldots = P/N$ as a hyperparameter that specifies the number, although $P$ was already used in the main text as the total number of predicates (i.e., $\sum_n P/n$).

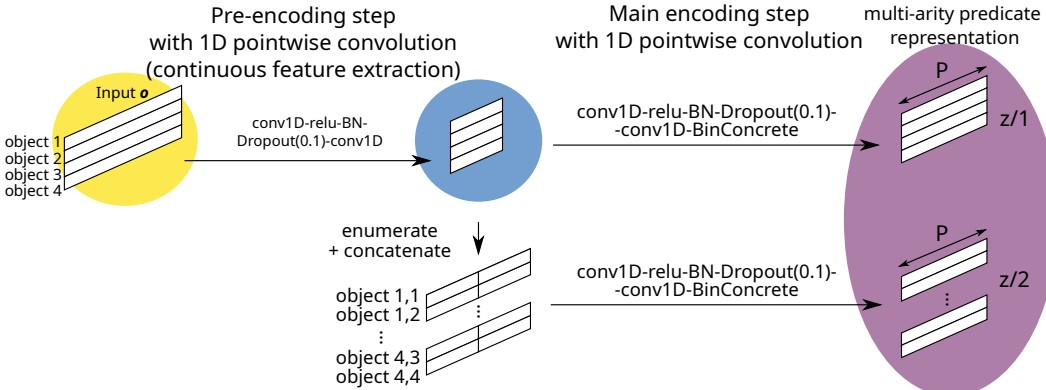

Figure 13: The encoder has two stages. The first stage extracts a compact but still continuous representation of each object. From this compressed representation, the second stage identifies properties of objects as well as relationships/predicates between objects.

### B.2.2 PARAMS

PARAMS consists of multiple NLM layers, as depicted in Fig. 14. All hidden activations are ReLU, and the width $Q$ of PARAMS is tuned by the GA tuner.

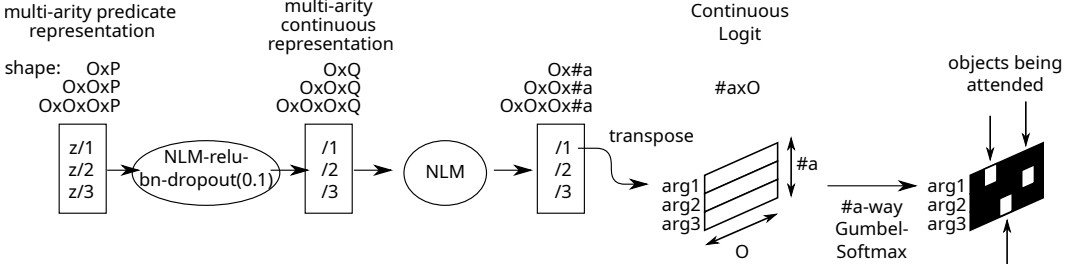

Figure 14: PARAMS network using NLMs.

### B.2.3 ACTION, APPLY / REGRESS

Since ACTION takes parameter-bounded representations $(z_{\dagger}^{i,0}, z_{\dagger}^{i,1})$ whose objects are already selected and appropriately reordered by BIND, we simply apply an MLP whose last layer has an output size $A$, and is activated by Gumbel-Softmax. This results in selecting one action, represented by a one-hot vector. Before applying the MLP, we should flatten and concatenate the tuple of vectors, each of size $\#a \times \overset{n}{\ldots} \times \#a \times P$. The depth and the hidden layer width of the MLP is automatically tuned.

For APPLY and REGRESS, we similarly flatten the input and directly apply the BTL structure described in the earlier sections. The EFFECT network in $\tilde{z}^{i,1} = \text{BC}(\text{BN}(z^{i,0}) + \text{EFFECT}(a^i))$ is a single linear layer without bias combined with a batch normalization. In other words, with the weight matrix $W$,

$$\tilde{z}^{i,1} = \text{BC}(\text{BN}(z^{i,0}) + \text{BN}(W a^i)).$$

### B.2.4 DECODER

The decoder consists of NLMs followed by a post-decoder that shares the same width and depth as the pre-encoder. In Blocksworld, we additionally made the decoder a Bayesian Neural Network to absorb the uncertainty in the reconstruction. Details are available in (Sec. D.2).

### B.3 HYPERPARAMETER TUNER

The tuning system assumes that the hyperparameter configuration is a vector/dict of categorical/finite candidates. The tuner is a textbook integer GA with uniform crossover and point mutation. Assume that a hyperparameter configuration is represented by $H$ values. A new configuration $p = \{p_1, \cdots, p_H\}$ is created from two parents $q = \{q_1, \cdots, q_H\}$ and $r = \{r_1, \cdots, r_H\}$ by $\forall i; p_i = \text{RandomChoice}(q_i, r_i)$, and then a single value is randomly mutated, i.e., for $m = \text{RandomChoice}(1..H)$, $p_m \leftarrow \text{RandomChoice}(\text{ValidValuesOf}(p_m) \setminus \{p_m\})$. It stores all evaluated configurations and never re-evaluates the same configuration.

In the beginning of the tuning process, it bootstraps the initial population with a certain number of configurations. New configurations are evaluated by the validation loss, then pushed in a sorted list. A certain number of best-performing configurations in the list are considered as a "live" population. In each iteration, it selects two parents from the live population by inverse weighted random sampling of the score, preferring the smaller validation losses but also occasionally selecting a second-tier parent. Non-performing configurations will "die" by being pushed away from the top as the algorithm finds better configurations. The evaluation and insertion to the queue is asynchronous and all processes can run in parallel.

## C  TRAINING DETAILS AND HYPERPARAMETERS

We trained multiple network configurations on a distributed compute cluster equipped with Tesla K80 and Xeon E5-2600 v4, which is a rather old hardware. The list of hyperparameters are shown in Table 3.

In all experiments, we used the total of 5000 state transitions (10000 states) from the training environments. The details of data collection for each domain is available in the later sections. This dataset is divided into training/validation/test sets (90%:5%:5%). The Genetic Algorithm hyperparameter tuner uses the validation loss as the evaluation metric.

We set a limit of 1500 total runs for each environment, with 100 initial population, and ran maximum 100 processes in parallel for each environment. As an additional trick, to avoid testing unpromising candidates (e.g., those with diverging loss), the epoch parameter is forced to be 50 in the first 100 configurations and the runs finish quickly. The rest of the runs use these initial populations as the parents, but replaces the epoch with an appropriate value selected from the candidates.

| | |
|---|---|
| Training parameters | |
| Optimizer | Rectified Adam, RMSProp, Nadam, Adam |
| Epochs | 100, 333, 1000 |
| Batch size | 100, 333, 1000 |
| Learning rate | $10^{-2}, \ldots, 10^{-5}$ |
| Gradient norm clipping | 0.1, 1.0, 10 |
| Initial annealing temperature $\tau_{\max}$ | 10, 5, 2 |
| Final annealing temperature $\tau_{\min}$ | 0.2, 0.5, 0.7 |
| Coordinate loss amplifier | 1, 10, 100 |
| Network shape parameters | |
| #p | 1, 2, 3 |
| #a | 1, 2, 3 |
| P | 10, 33, 100, 333, 1000 |
| A | 10, 33, 100, 333, 1000 |
| Pre-encoder/decoder output dimension | 10, 33, 100, 333, 1000 |
| Encoder hidden layer width | 10, 33, 100, 333, 1000 |
| Encoder depth | 2 |
| Decoder hidden layer width | 10, 33, 100, 333, 1000 |
| Decoder depth | 2 |
| Params/Action/Apply/Regress hidden layer width | 10, 33, 100, 333, 1000 |
| Params/Action/Apply/Regress depth | 1, 2, 3 |
| Params/Action/Apply/Regress hidden activation | ReLU, tanh |
| Regularization Parameters | |
| Latent L1 regularization $\alpha$ | 0.0, 0.01, 0.1, 0.2 |
| KL divergence coefficient $\beta$ | 0.0, 0.1, 0.3 |
| Latent dynamics loss $\gamma$ | 0.1, 0.2, 0.3, 0.5, 0.8 |
| Warmup parameters | |
| Latent L1 regularization $d_\alpha$ | 0.0, 0.01, 0.1, 0.2, 0.3, 0.5, 0.8 |
| Main reconstruction loss $d_{\mathrm{rec}}$ | 0.0, 0.01, 0.1, 0.2, 0.3, 0.5, 0.8 |
| Dynamics-based reconstruction loss $d_{\mathrm{dyn}}$ | 0.0, 0.01, 0.1, 0.2, 0.3, 0.5, 0.8 |
| Latent dynamics loss $d_\gamma$ | 0.0, 0.01, 0.1, 0.2, 0.3, 0.5, 0.8 |
| Early stopping | |
| Explosion detection | $10\times$ the loss value at the epoch 0 |

Table 3: List of hyperparameters tuned by the Genetic Algorithm.

### C.1  REPRODUCIBILITY, TRAINING CURVES ON 3 RUNS

In Fig. 15, we show 3 training runs on each environment with the same hyperparameter configuration found by our tuner. It reproduces an almost identical, stable curves across 3 runs.

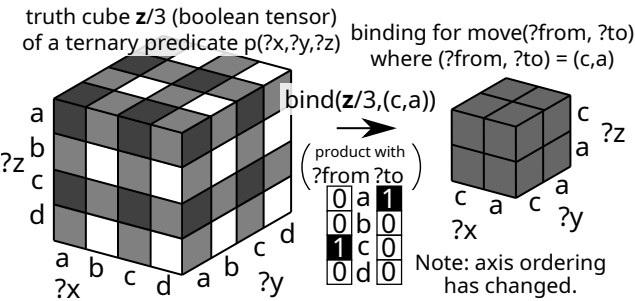

Figure 15: Training curves of 3 runs on the same optimized parameter found by the parameter tuner. The plots are produced by Tensorboard. We can see that the training is quite stable and the curves are similar.

# D  DOMAIN-WISE DETAILS

## D.1  SOKOBAN

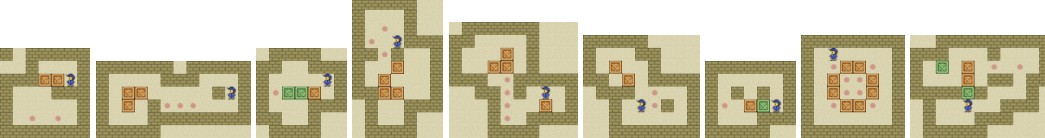

Figure 16: Visualized sokoban problems. The first 5 are used for the training, and the rest are used for evaluation. Pink dots depict the goals that the player pushes the blocks onto. Green boxes are already on one of the goals.

We generated 10000 transitions of each training problem using Dijkstra search from the initial state. We shuffled the 50000 transitions, subsampled 5000 transitions out of 50000, then stored them in a single archive. The rendering and other data are obtained from PDDLGym library Silver & Chitnis (2020). Each tile is resized into 16x16 pixels, and the tile ordering is also shuffled.

In order to make the training data dimension consistent and make it convenient for GPU-based training, we performed a so-called *Random Object Masking* which removes a certain number of objects from each state via random selection. The idea is similar to masking-based image augmentation commonly used in the computer vision literature, but the purpose is different from those: Ours has more emphasis on having the consistent number of objects in each state. For example, Sokoban training problem 0 (leftmost in Fig. 16) has 49 tiles, while the problem 1 (the second picture) has 72 tiles. In the combined archive, the number of objects is set to the smallest among the problems.

We also removed certain tiles that cannot be reached by the player. For example, in problem 0, the three floors on the top left corner cannot be reached by the player. Similarly in problem 1, the bottom right corner is not reachable by the player. We performed a simple custom reachability analysis using the meta-information obtained from PDDLGym library. This helped reducing the dataset size and thus the training.

Finally, during the dataset merging, we accounted for the potential location bias caused by the map size difference. For example, if we preserved the original $x, y$ locations, the tiles tend to be biased around $0, 0$ and the location around $(x, y) = (12, 12)$ (by tiles) is never occupied by any tile. To address the issue, for each state pair, we relocated the entire environment by a value selected uniformly randomly from a certain width. The width is decided from the maximum dimension of all training problems, i.e., 12x12. For example, a state pair in problem 4 (which has a 10x9 map dimension) will be shifted by a random value between $0..(12 - 10)$ in $x$-axis, and $0..(12 - 9)$ in $y$-axis.

## D.2  BLOCKSWORLD

We generated 5000 random state transitions using Photorealistic-Blocksworld dataset Asai (2018), which in turn is based on CLEVR Johnson et al. (2017) dataset generator. It uses Blender 3D rendering engine to produce realistic effects such as metallic reflection, surface materials and shadows. The objects are extracted from the information available from the generator metadata. We cropped the image region suggested by the metadata, resized them into 32x32x3 image patches and stored them in an archive. The size of the objects reported by the metadata may vary and is noisy due to the camera jitter, object location, and the heuristics used for extracting the objects. This is a case even for the objects that are not moved by the random actions.

Each transition is generated by sampling the current state and then randomly moving a block. To sample a state, we first generate a set of block configurations (color, material, shape, size), then placing them randomly on a straight line in the 3D environment without collisions. When we move a block, we select a set of blocks of which nothing else is on top, choose one randomly, pick a new horizontal location and place it at the lowest non-colliding height. We ensure that the block is always moved, i.e., not stacked on top of the same block, and not on the floor if it is already on the floor.

In Blocksworld, we noticed that a same conceptual symbolic action (such as move) may have a non-deterministic outcome in the image space while each individual concept is discrete and deterministic.

For example, move may relocate a block onto a floor, but the resulting position is chosen randomly during the dataset generation, i.e., it can be placed anywhere on the floor.

To model this uncertainty in our framework, we used a Bayesian Neural Network layers in the decoder for Blocksworld domain. The final output $o \in \mathbb{R}^{O \times F}$ is produced by two NLMs of output feature size $F$, each producing the mean and the variance vectors $\mu, \epsilon \in \mathbb{R}^{O \times F}$, and the output reconstruction is generated by the random sampling: $o = \mu + \sigma \cdot \epsilon$, where $\epsilon$ is a random noise vector following normal distribution $\mathcal{N}(0, 1)$. During the testing (e.g., visualization), the random sampling is disabled and we use the value $\mu$ for rendering.

### D.3 8-PUZZLE

The 8-puzzle domain generator is directly included in the Latplan code base. It uses MNIST dataset for its tiles and the tile size is set to 16. Similarly, we generated 5000 random state transitions using the generator.

## E PDDL GENERATION AND PLANNING EXPERIMENTS

### E.1 EXAMPLE OUTPUT: 8-PUZZLE

