# OpenReview forum: "Generating Plannable Lifted Action Models for Visually Generated Logical Predicates"
_ICLR.cc/2021/Conference — Reject_

### Official Review · AnonReviewer2 · 2020-10-26
**A solid contribution, but with sparse results**

**Rating:** 6
**Confidence:** 3

**Review:**

This work presents FOSAE++, an end-to-end system capable of producing "lifted" action models provided only bounding box annotations of image pairs before and after an unknown action is executed. Building on recent work in the space, the primary contribution of this work is to generate PDDL action rules. To accomplish this, the authors introduce novel 'params' function that use the Gumbel-Softmax function to implement a differentiable mechanism for selecting which entities are relevant to the current action and feeds them into the new 'bind' and 'unbind' functions that select those elements in the tensor predicting their relevance. Overall, this work is a meaningful contribution in the direction of generated lifted action models without direct labeled data.

The biggest flaw in the paper is the rather sparse results section. Additionally, I do somewhat take issue with the statement in 4.2 that "due to the time constraint" planning experiments beyond the 8-Puzzle domain. Though I appreciate the author's candor in this regard, at the moment the paper is rather weakened by the lack of inclusion of additional planning experiments, particularly since the 8-Puzzle domain is arguably the easiest domain from which one might generated a lifted action representation, due to the black background surrounding the digits. Seeing planning experiments in the Sokoban domain would greatly strengthen the paper. The reconstructed Sokoban environments in Figure 4 look rather good, though the slightly shifted tiles in the reconstructed scene results in black lines/gaps between the cells, raising questions about the ability of the system to perform planning in these domains.

Relatedly, though I appreciate that the authors are space-constrained at the moment, the results section (and in particular the planning section) is quite short and lacking in detail. More detail, including discussion of the limitations of this approach, would strengthen the paper.

The Appendix is incredibly thorough and a welcome addition to the paper. It provides helpful additional content that, while not necessary for understanding the paper, aid in understanding and implementation.

Smaller comments:
- The caption for Fig. 2 should be extended or more annotations should be added to the figure itself. Right now, it is only clear from the body text how these components are used or where these models were introduced in other papers. This change would help clarity.
- A passage addressing the limitations of the proposed system would be a welcome addition as well. In particular, there seems to be a general assumption that only the regions within the provided bounding boxes will change after an action is executed, something this is not generally true (nor is true in the Photo-realistic Blocksworld domain, in which shadows can change outside the bounding boxes).
- The caption in Fig. 3 is (I think) incorrect: the final note should read that `move` was simplified to `?from, ?to`.
- Fig. 12 in the appendix is a duplicate of another figure earlier in the paper. It should be replaced with the _actual_ figure before publication.

---

### Official Review · AnonReviewer1 · 2020-10-27
**Review of Generating Plannable Lifted Action Models for Visually Generated Logical Predicates**

**Rating:** 5
**Confidence:** 1

**Review:**

## Summary

This paper builds on Latplan (Asai & Fukunaga 2018) and proposes FOSAE++, a lifted action model expressed in First-Order Logic (FOL), which differs from object-centric representation by being generalized over objects and environments. According to the authors, FOSAE++ is the first system that :
* uses a white-box action model which is trivially convertible to STRIPS formalism,
* is invariant to the number/order of objects,
* uses unsupervised generation of multi-arity predicate symbols.

FOSAE++ is demonstrated on three environments:
* Photo-realistic Blocksworld
* MNIST 8-puzzle
* Sokoban
In each environment the input is the set objects obtained using a domain-specific segmentation of the image.

## Reason for score

I'm sorry but by expertise in the domain of this paper is way too limited to be able to do a fair assessment. From the introduction I was not able to clearly understand the contributions of the paper. This is not necessarily due to the introduction, but rather to my lack of expertise in the domain. Many of the concepts are alien to me (PDDL, classical planning, lifted action models, object-centric representations, STRIPS).

That being said, I did find the introduction particularly opaque and it was difficult for me to clearly identify the contributions of the paper. I was not able to make sense of Table 1 and believe the authors could have done a better job of introducing the different aspects that distinguish their methods from previous work.

I did give a good read to the rest of the paper but, again, am unable to comment deeply on the content. I did find the figures particularly dense and difficult to parse. I was also surprised that more than 2 pages were dedicated to background and previous work. Despite that long exposition, I would have to read a lot of the referenced work in order to understand these preliminaries. Again, this may be due to my lack of expertise however I can't help but think that the authors could have done a better job at bringing me up to speed.

Also, I'm a bit surprised that the authors claim to have achieved "unsupervised end-to-end neural system that generates a
compact discrete state transition model (dynamics / action model) from raw visual observations." given that they rely on domain-specific segmentation code to extract objects, and that their technique only operate on these objects. They claim that "in principle [the domain specific code] can be replaced by the output of object-detectors such as YOLO" but I find this unconvincing.

## Conclusion

Given my very limited understanding, I would only consider my rating as an educated guess based on a small number of details I was able to evaluate.

---

### Official Review · AnonReviewer4 · 2020-10-28
**Inspiring paper on a less studied problem**

**Rating:** 6
**Confidence:** 4

**Review:**

This paper addresses the problem of learning dynamics model directly from raw sensory inputs. The authors propose an unsupervised end-to-end model that can perform high-level tasks planning on raw observations. This work extends Asai et al. 2020, 2019 etc, and with improved symbol generation and lifted PDDL. The authors follow the experimental setup as seen in prior work, where three artificial environments (blocksworld, MNIST 8-puzzle, and sokoban) are used for planning.

Pros.
+ interesting and important problem of high-level tasks planning with potentially practical usage
+ detailed analysis of experiments in the supplementary
+ the model seems to be working really well without much supervision
+ training details are provided for re-implementation

Cons.
- The notations used throughout the paper can be a little bit confusing. Readers who have not read supplementary and prior works may find it difficult to understand.
- Experiments are limited to relatively simple environments. It would be interesting to see how the proposed model can be applied to the more realistic scenarios as seen in Konidaris et al.

Overall the reviewer thinks this paper addresses an important yet less studied topic, and will likely inspire follow-up works towards this direction. While the paper has some problems in terms of presentation, the reviewer recommends its acceptance.

---

### Decision · Program_Chairs · 2021-01-07
**Final Decision**

**Decision:**

Reject

**Comment:**

This paper described a system for deriving PDDL (Planning Domain Description Language) operator descriptions from unlabeled visual image pairs.  The goal is to construct STRIPS-like descriptions with preconditions, add lists and delete lists for operators that can explain the state transitions seem in the image pairs.  This is combined with a neural form of inductive logic programming (ILP) which historically performs a similar task from logical descriptions rather than images.

While this topic is appropriate for ICLR, the work is fairly incremental and the experiments are limited to the 8-puzzle which, according to a reviewer, is the easiest of the tasks.  In spite of boarder line scores, no rebuttal was provided.  So my recommendation is to not accept the paper.